# Movie viewing elicits rich and reliable brain state dynamics

Johan N. van der Meer [1✉], Michael Breakspear [2,3], Luke J. Chang [4], Saurabh Sonkusare[1] & Luca Cocchi [1✉]

Adaptive brain function requires that sensory impressions of the social and natural milieu are dynamically incorporated into intrinsic brain activity. While dynamic switches between brain states have been well characterised in resting state acquisitions, the remodelling of these state transitions by engagement in naturalistic stimuli remains poorly understood. Here, we show that the temporal dynamics of brain states, as measured in fMRI, are reshaped from predominantly bistable transitions between two relatively indistinct states at rest, toward a sequence of well-defined functional states during movie viewing whose transitions are temporally aligned to specific features of the movie. The expression of these brain states covaries with different physiological states and reflects subjectively rated engagement in the movie. In sum, a data-driven decoding of brain states reveals the distinct reshaping of functional network expression and reliable state transitions that accompany the switch from resting state to perceptual immersion in an ecologically valid sensory experience.

[1] Program of Mental Health, QIMR Berghofer Medical Research Institute, Brisbane, 300 Herston Road, Brisbane 4006 QLD, Australia. [2] School of Psychology, Faculty of Science, University of Newcastle, University Drive, Callaghan, NSW, Australia. [3] Discipline of Psychiatry, Faculty of Health and Medicine, University of Newcastle, University Drive, Callaghan, NSW, Australia. [4] Department of Psychological and Brain Sciences, Dartmouth College, Hanover NH 03755 NH, USA. ✉email: Johan.vanderMeer@qimrberghofer.edu.au; luca.cocchi@qimrberghofer.edu.au

A key function of the brain is to integrate dynamic sensory inputs with internal intentions, motivations, and predictions about the world[1]. This ability is required for adaptive navigation of the world[2]. The experience of watching a movie relies upon the dynamic processing of its audiovisual content to form and update the impressions and expectations of the next scene. Our understanding of the coordinated patterns of brain activity that support the evaluation of sensory information has been advanced by dynamic, multivariate, and network-based analyses of functional neuroimaging data[3,4]. However, capturing brain state dynamics remains a challenging endeavour and their involvement in perception, evaluation, and action remains unclear.

The brain manifests coordinated changes of activity across multiple cortical regions, even in the absence of external tasks[5,6]. Dynamic patterns of functional brain connectivity at rest appear to reflect task-based phenotypes, including processing speed and fluid intelligence[7,8]. Dynamic jumps between discrete brain states can be modelled using the hidden Markov model (HMM), an analytical framework that posits the existence of distinct states, whose sequential expression yields observed functional imaging data[9]. Here, HMM states define spatial patterns of fMRI signal magnitude that recur sporadically in time. The recent application of the HMM to resting-state activity has shown that such discrete, transitory brain states are linked to genetics and behavioural factors, including intelligence and personality[5]. Although the HMM is a data-driven approach, the sequential expression of discrete states links naturally to theoretical mechanisms of brain dynamics, including metastability[10] and multistability[11,12].

The variability of resting-state neural dynamics across participants and their unconstrained nature limits the ability to make direct inferences about the behavioural relevance of brain state dynamics. Furthermore, common methods adopted to infer macroscopic dynamics, including dynamic functional connectivity, are prone to non-neural confounds such as head motion, cardiac noise, and respiratory artefacts[13]. Thus, it remains unclear how brain functions rely on major dynamic reconfigurations of whole-brain functional patterns[14,15]. Conventional task designs, which typically comprise discretely presented, abstract stimuli also impede the assessment of associations between ongoing stimulus processing and its evaluation such as the engagement and interest in the narrative of a story or a movie[16].

Naturalistic stimuli, such as movies[3] and spoken narratives[17], offer the constraint and replicability that resting state acquisitions lack while adding greater ecological validity than traditional task designs[18]. Recent analyses of movie viewing fMRI data using the HMM have revealed a hierarchy of timescales, with more frequent state transitions in sensory cortex hierarchically nested within progressively slower transitions in heteromodal regions[19]. Such multiscale dynamics mirror the statistics of the natural world[20] and suggest a remodelling of intrinsic correlations so that their complexity more closely matches the statistical structure of naturalistic perceptual streams[4,21].

Comparing unconstrained resting-state acquisitions to movie viewing has the potential to characterise this remodelling process, and hence to investigate the functional significance of transitory brain states. Using the HMM approach[8], we mapped brain states in fMRI data associated with both resting state and movie viewing on two occasions, three months apart. The validity of the inferred brain states was assessed using cross-session comparisons, movie annotations, the Neurosynth database[22], and concurrently recorded physiological indices including heart rate (HR) and pupil diameter (PD). We also investigated if metrics related to brain state dynamics during movie watching are correlated with the individual subjective immersion in the movie. This unique design allowed disambiguation of reliable, stimulus-driven states from endogenous brain dynamics. movie viewing induces the reshaping of spontaneous brain dynamics into a reliable sequence of states whose occurrence are temporally aligned to specific features of the movie and reflect subjective engagement in the movie.

## Results

**Brain states differently expressed in movie compared to rest.** Functional magnetic resonance imaging (fMRI) data were analyzed for 18 healthy participants who were scanned during 8 min of resting-state followed by 20 min of movie viewing. HR and PD were also recorded. All participants completed a questionnaire immediately following the first neuroimaging session. Fourteen of these participants repeated this experimental session after 3 months.

Brain states occurring at rest and during movie viewing were estimated using the HMM, a method which posits that the observed data arise from a small number of hidden states and their transitions[5]. To allow for a direct comparison between the states during the different experimental conditions (baseline rest and movie viewing, plus 3-month follow-up rest and movie viewing), we estimated brain states using concatenated time series of 14 participants who completed both experimental sessions (Methods). This allowed obtaining a group estimation of brain states for each experimental condition and session. The inversion of the HMM from these data yielded ten distinct states (Fig. 1). Confirmatory analyses were performed on 8 min of rest and 8 min of movie data, with HMM inversions performed on concatenated data as well as performed separately (i.e., movie and rest independently; Supplementary Figs 2–5). To understand the functional expression of these states, we coded their respective loadings onto each of 14 widely studied canonical brain networks (BNs)[23] (Supplementary Fig. 1). The expression of network activity was normalised so that zero corresponds to the average activity of that network across the movies and rest periods. The variability was also scaled according to that network's standard deviation, allowing insights into the balanced representation of changes in fMRI signal magnitude across canonical BNs in time. Each network was normalised separately (Methods). Each brain state represents structured, recurring patterns of activity, loading distinctly across these fourteen canonical BNs at any specific timepoint.

Brain states are characterised (see Fig. 1) by their distinct fMRI signal loadings onto the 14 canonical BNs (Supplementary Fig. 1). State 1 is defined by high fMRI signal in most networks. States 5 and 9 show a relatively uniform fMRI signal across networks, whereas State 7 displays a low projection onto all networks (except for the primary visual network). The remaining brain states (2–4, 6, 8, and 10) show idiosyncratic fMRI signal across the 14 BNs. The fMRI signals defining these brain states load preferentially on one or more specific functional networks such as those supporting language (state 3), visual-auditory stimuli processing (state 2, 4, and 10), and interoception (states 6 and 8); see Supplementary Fig. 3 for confirmatory analyses on HMM inversions on 8 min of rest and movie data separately and Supplementary Fig. 4 for results on HMM inversions on 8 min each of concatenated data.

**States are consistent over participants during movie watching.** We next assessed the degree of inter-subject consistency in the expression of these ten states while participants watched the movie or underwent the resting-state condition. Using a reverse-inference approach[3,24], we investigated whether fluctuations in the level of between-subject consistency corresponded with the occurrence of

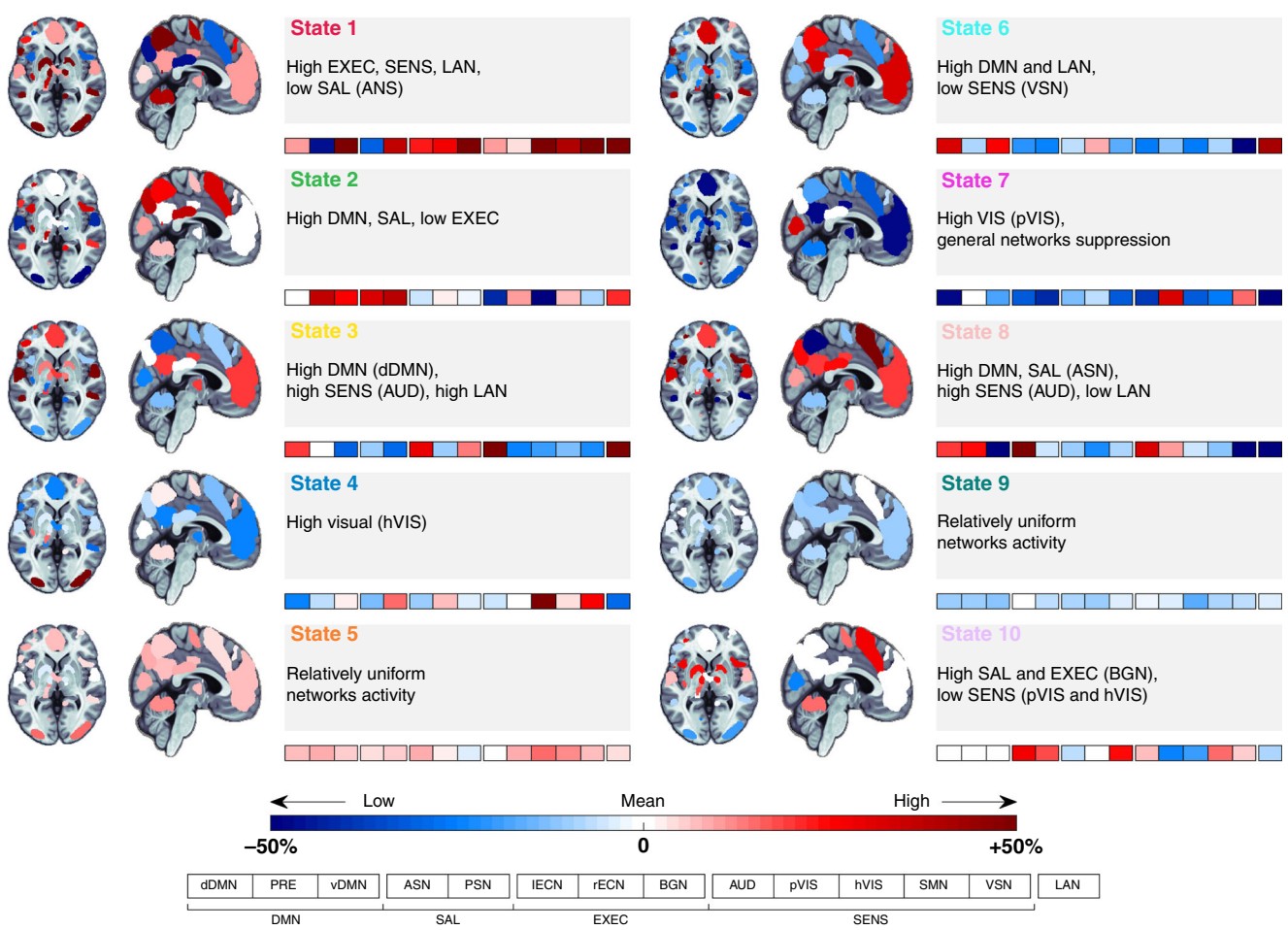

**Fig. 1 fMRI signal profiles for each brain state occurring during resting state and movie viewing.** The relative weighting of each brain state onto each of the 14 canonical networks considered: dorsal and ventral Default Mode Networks (dDMN and vDMN), Precuneus, Anterior Salience Network (ASN), Posterior Salience Network (PSN), Left and Right Executive Control Networks (lECN and rECN), Basal Ganglia Network (BGN), Auditory Network (AUD), Primary Visual Network (pVIS), High Visual Network (hVIS), Sensorimotor Network (SMN), Visuospatial Network (VSN), and Language Network (LAN). These are divided into four main groups: DMN (Default Mode Network); SAL (Salience Network); EXEC (Executive Network); and SENS (Sensory Network). The colour coding associated to each brain state label is maintained throughout the manuscript. The Blue-Red colour bar indicates the relative loading to the average activity across both recordings used to infer the HMM model.

particular events of the movie. Specifically, we counted how many participants visited a particular brain state in a small window around each time point and aligned these to independent annotations of the movie's structure and narrative (Table 1).

As anticipated, movie viewing was associated with greater consistency across participants, relative to rest (Fig. 2; also Supplementary Figs 3, 4, and 6). Moreover, the co-occurrence of brain states across participants reached the highest levels during specific movie events (Table 1), with complete consistency observed during the viewing of 11 different scenes in either session A or session B (Fig. 2). For example, in the third act of the movie (from 14 m:14 s to 14 m:40 s), brain state 6 (high DMN and language network expression) was visited by all participants during the first viewing session and 12 participants in the second session (86% consistency). States 5, 9, and 10 were not often present during these consistent times. In contrast, high levels of between-subject consistency did not occur during the resting-state acquisition (Fig. 2).

We also assessed the inter-session consistency by calculating the Jaccard index over state visits across session A and session B, averaged over brain states and participants (see Methods). The occurrence of brain states was significantly more consistent during movie viewing than rest (average Jaccard overlap of 0.18

(+/−0.04) in movies and 0.08 (+/−0.07) at rest, $p = 0.0020$). This finding is in line with the higher between-session consistency within each participant during movie compared to rest (Supplementary Fig. 6). These results highlight the relatively high (across-session and between-subject) consistency of brain state dynamics using a naturalistic stimulus, compared to the unconstrained resting-state condition. Formal comparison between inter-session consistency (for each subject separately) and inter-subject consistency (average inter-subject consistency) showed higher inter-session consistency (paired $t$-test, $t_{26} = 2.85$, $p = 0.008$; see also Supplementary Fig. 7). This result suggests the existence of movie-related participant-specific neural signatures.

In contrast, resting-state acquisitions were mainly dominated by bistable transitions between two states (5 and 9), although their temporal transitions were poorly aligned between participants. The higher reliability in the movie condition was linked to richer brain state transitions between a larger number of states compared to these bistable dynamics observed at rest (Supplementary Fig. 6).

**Brain states map to distinct behavioural profiles.** To quantify the functional relevance of the inferred states, we used association mapping between their spatial expression and Neurosynth topics

**Table 1 Annotation of the movie scenes.**

| State Interval (start) | State Interval (end) | | Consistency session A (%) | Consistency session B (%) | Annotation time | Description |
|---|---|---|---|---|---|---|
| 5:04 | 5:19 | State 3 | 79 | 100 | 5:03 | Turning point: The protagonist (Will) is told by tattooed man: "You just spat on the showman from The Butterfly Circus. That was Mendez..." |
| | | | | | 5:14 | He laughs cruelly at Will and we observe Will's reaction in close-up as he realises his predicament. |
| 5:19 | 5:37 | State 2 | 100 | 100 | – | (Not in Annotations) Will's former boss introduces him as 'The Limbless Man' and raises the curtain. But Will is gone. |
| 7:00 | 7:27 | State 3 | 100 | 86 | 7:07 | Obstacle: Will discovers that he cannot attain his goal of joining a "fancy show" because The Butterfly Circus does not have a sideshow and Will has no skills to perform his own act. |
| 8:00 | 8:13 | State 4 | 71 | 100 | | (Not in Annotations) Circus acts of Acrobat and Strongest Man. Will looks at the Audience |
| 8:15 | 8:33 | State 2 | 86 | 100 | | (Not in Annotations) Scenes of circus performances: The Flame Breather and The Houdini Water Tank Escape |
| 9:21 | 9:39 | State 1 | 100 | 93 | 09:38 | Obstacle: An African-American boy admires the Strongman's muscles and asks Will if he is also in the Butterfly Circus. |
| | | | | | 09:40 | Will responds to the boy sadly, "No, not exactly". |
| 12:12 | 12:39 | State 4 | 100 | 86 | | (Not in Annotations) Past scenes of difficult past of other circus members, contrasted with how they are now |
| 14:14 | 14:40 | State 6 | 100 | 86 | 14:10 | Confronts Obstacle: Will falls while trying to cross the river. As Will shouts for help his circus friends in the distance cannot hear him over the sound of the river. |
| | | | | | 14:20 | Mendez walks by. Will asks Mendez for help but Mendez says, "I think you'll manage." |
| 15:44 | 16:08 | State 6 | 100 | 76 | 15:34 | Confronts Obstacle: Will falls off the log into deep water. |
| | | | | | 15:46 | The circus folk notice Will is missing and search frantically for him in the water. |
| 16:32 | 16:39 | State 4 | 100 | 93 | 16:20 | Overcomes Obstacle: Will overcomes his external obstacle and bobs to the surface: "Look! I can swim!" |
| 16:43 | 17:03 | State 7 | 100 | 86 | 17:05 | Climax: Mendez tells the audience that Will is climbing 50 feet into the air and he will leap from a high platform |

These annotations highlight the very high inter-subject consistency either in session A or session B. The time corresponding to the movie is given in minutes:seconds. Further details about the movie are available in Supplementary Note 1. The brief descriptions provided here match independent movie annotations; i.e., Turning point, Development, Obstacle, or Climax. The full details and annotations are provided in Supplementary Table 1.

(reverse inference[22,24]; see Methods). This approach yields a functional profile of the ten brain states across sixteen selected topics: anxiety, language, negative, positive, outside, task switching, inhibition, conflict, feedback, pain, somatosensory, sensorimotor, music, auditory, emotion, and face perception (Fig. 3). Brain states showed distinct functional signatures. States 4 and 7 have patterns that map onto task switching and sensorimotor functions; State 3 has a strong association with language, emotion, and auditory processing; State 6 has an association with emotion processing with both positive and negative loading; and State 10 is linked to sensorimotor processes, pain, and inhibition.

**Brain states link to autonomic indices and movie annotations.** Physiological changes are integral to emotional experiences[25], and naturalistic stimuli are known to evoke reliable physiological changes[26]. We, therefore, assessed whether the occurrence of a given brain state corresponded to distinct electrophysiological signatures of autonomic function associated with changes in sensory inputs and level of arousal, namely HR and PD[27,28]. We observed several significant associations between the occurrence of brain states and fluctuations in both HR and PD (Table 2). For example, lower HR was associated with the occurrence of brain state 3 (low dDMN and language network activity), which were in turn characterised by a neutral (i.e. low anxiety and pain) functional profile (Fig. 3). This result supports the link between changes in movie-induced arousal states and the emergence of brain states. Smaller PD occurred during the expression of brain state 4, which was characterised by high visual network activity linked to face perception. We found a strong negative correlation between PD and scene total luminance ($r = -0.69$, $p = 10^{-75}$), supporting the notion that changes in PD are linked to changes in sensory inputs. However, larger PD coincided with the occurrence of brain states 1 (high executive, sensory and language) and

2 (high DMN, salience but low executive). In line with high DMN activity, state 2 is functionally linked to high anxiety[29] and pain[30]. PD may therefore also link, at least to some extent, to transient interoceptive mechanisms. Previous work has shown evoked PD in the absence of visual stimuli may reflect higher-order cognitive processes, such as updated sensory expectations following surprise[31], consistent with the association between larger PD and State 1.

The transitory nature and switching of brain states is likely linked to the unfolding of the movie. To test this hypothesis, we calculated the overlap between states and the following movie annotations: (i) presence of faces with positive or negative expressions, (ii) presence of positive or negative valence of scenes, (iii) use of language, and (iv) changepoints (i.e., scene changes). We observed strong, statistically significant associations ($p_{FWE} < 0.05$) between the occurrence of brain state 1 and the presence of positive scenes and facial expressions (Table 3), which were in turn associated with low anxiety and larger PD (Table 2). Brain state 6 was associated with the presence of negative facial expressions and, in turn, with high anxiety. State 3, which weighted heavily onto the language networks (Fig. 1), was expressed during scenes involving spoken dialogue. Finally, changepoints were linked to brain states 2 and 7. These two brain states were linked to the NeuroSynth constructs of scene switching (Fig. 3). Brain state 2 linked with inhibition in high arousal (large PD and high negative emotions and anxiety).

In principle, it would be interesting to understand whether temporal disruptions in the movie dynamics (e.g., scene changes) coincide with temporal discontinuities in brain states (i.e. state transitions). However, movie change-points tend to be clustered together, frequently occurring several times a second, interspersed by relatively long continuous scenes (Supplementary Fig. 8). Thus, the relatively slow temporal resolution of our BOLD-derived brain

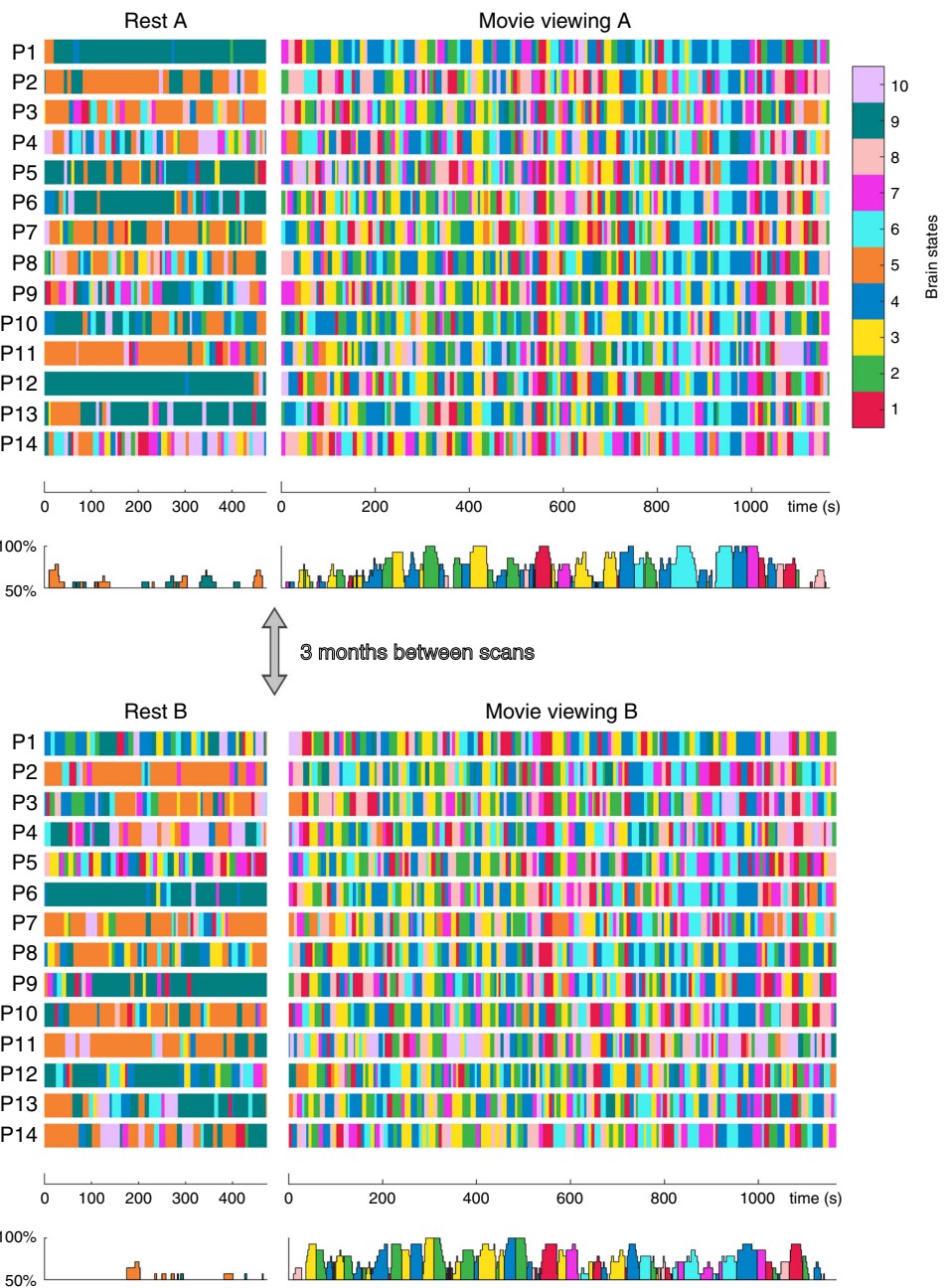

**Fig. 2 Brain state dynamics for rest and movie viewing, for each participant and session.** Brain states are colour coded according to the legend (top right, refer to Fig. 1 for brain state topologies). The temporal consistency across participants, which vary between 50% (chance for the resting state) and 100% (complete consistency), is also presented (bottom panel of each session). Across-participants and –sessions consistency are relatively low during pre-movie rest. Conversely, during movie viewing, across-participants and -sessions consistency are high. Specifically, across-participant consistency reaches 100% during the viewing of specific movie scenes (see Table 1 for details).

state transitions does not allow for a meaningful assessment of any putative temporal relationships. However, given the frequent occurrence of clustered, bursty dynamics reported in neurophysiological recordings[32], the use of rapidly sampled data modes (e.g., M/EEG) could address this question in future studies.

**Analyses of brain state dynamics: rest versus movie viewing.** We next quantified the dynamics of the ten brain states by assessing the following measures: (i) fractional occupancy (FO), defined as the proportion of total time spent in each given state; (ii) state dwell time, representing the total amount of time spent in each state, and (iii) state transition probabilities, representing

the likelihood of specific transitions between distinct brain states (Fig. 4; see Supplementary Figs. 3–5 for confirmatory analyses).

Relative to rest, movie viewing was characterised by significantly higher FO in brain states 1–4 and 6–8 (Fig. 4b). These states were characterised by high weightings in visual, auditory, and language networks (Figs. 1 and 3). The duration of the brain states was in between the annotations with relatively shorter duration (positive/negative faces: 7.4 +/− 5.6 s; positive/negative valence scenes: 8.2 +/− 7.4 s; language: 14.6 +/− 13.2 s; and changepoints: 4.0 +/− 1.9 s), and scene descriptions with relatively longer duration (Table 1). As noted above, the situation was the opposite in the resting state, with higher occupancy of brain

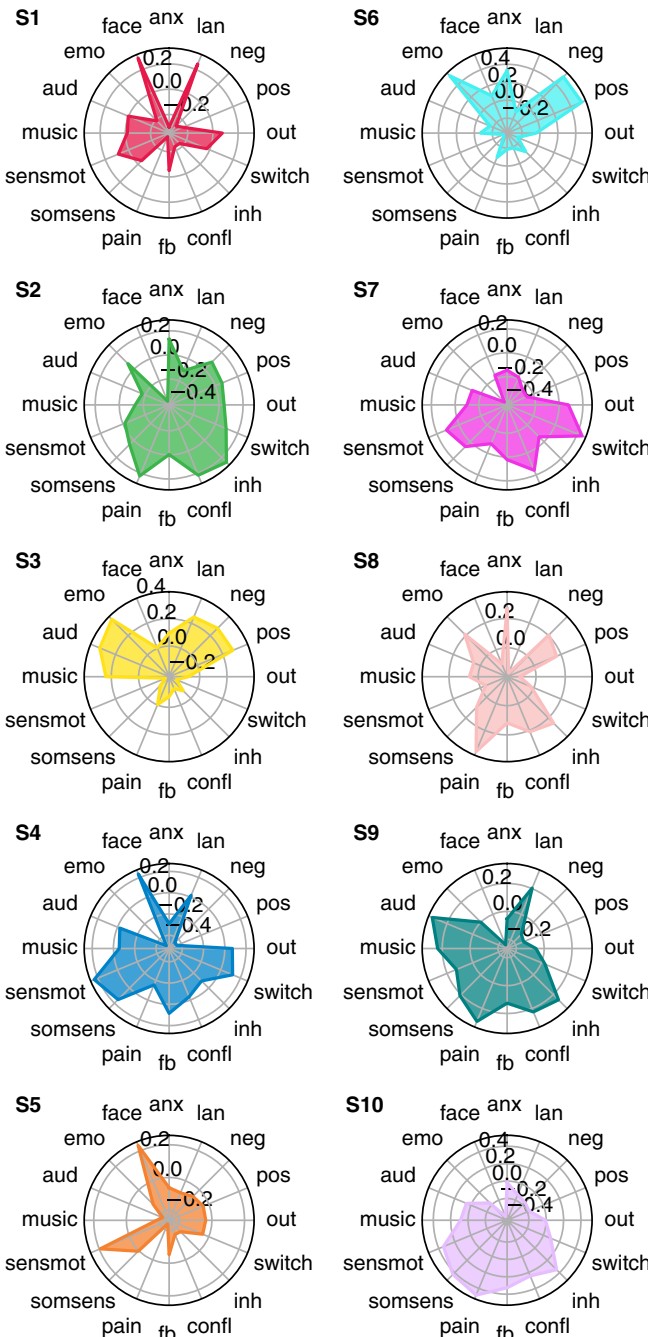

**Fig. 3 Probabilistic functional profile of brain states based on meta-analytic patterns of fMRI activity.** 'Topic maps' used to match the brain states were generated to catalogue associations between patterns of brain activity (fMRI) and topics in the scientific literature[22]. Scores represent the correlation between each brain state and Neurosynth topics. anx: anxiety; lan: language; neg: negative; pos: positive; out: outdoor; switch: task switching; inh: inhibition; confl: conflict; fb: feedback; somsens: somatosensory; senmot: sensorimotor; aud: auditory; emo: emotion; face: face perception.

**Table 2 State-specific deviations.**

|           | ΔHR (bpm) | ΔHR (*p* value) | ΔPD (a.u.) | ΔPD (*p* value) |
|-----------|-----------|-----------------|------------|-----------------|
| State 1   |           |                 | 45.7       | 0.014           |
| State 2   |           |                 | 90.3       | 0.023           |
| State 3   | −0.42     | 0.021           |            |                 |
| State 4   |           |                 | −11.4      | 0.001*          |

Deviations from their mean value across the entire movie are given in beats per minute (bpm) for heart rate (HR) and arbitrary units (a.u.) for pupil diameter (PD).
States 5 to 10 exhibited no significant deviations in HR or PD.
*$p_{FWE}$ < 0.05; one-sample two-sided *t*-test

overall reduction in time spent visiting each specific brain state. Also, the visited brain states were characterised by strong and specific patterns of fMRI signal in canonical functional networks. In other words, movie viewing induced a higher number of briefer brain state visits compared to rest. These results were replicated in the second Movie viewing and rest sessions (Supplementary Fig. 9; confirmatory analyses are presented in Supplementary Figs. 3 and 5).

Movie viewing hence imposes a faster overall turnaround of brain states, primarily due to less occupancy of brain states 5 and 9 that have longer dwell times in the resting state data. To further understand this observation, we calculated state transition probabilities (Fig. 5). Significant differences between brain state transitions at rest and during movie viewing were identified using Network-Based Statistics ($p_{FWE}$ < 0.05, Methods). These analyses revealed a significant increase in the likelihood of transitions from state 9 to state 5 in the resting state compared to movie viewing (Fig. 5e). The reverse contrast (movie > rest) revealed a higher number of transitions between a number of specific brain states associated with movie viewing, particularly among states 1, 2, 3, 7, and 8 (Fig. 5f). These findings were also replicated across the two scanning sessions (Supplementary Fig. 10).

We also explored quantitative similarities between the time-locked individual state-paths across participants—that is, the detailed sequence of states and the precise timing of individual transitions. Although such a sequence is a relatively complex path unfolding on a high-dimensional manifold, the (Jaccard) dissimilarity index for all subject pairs ranged from 0.72 to 0.88, a moderate level of agreement. That is, despite the misalignment in the precise timing of the state transition times, there is still a meaningful commonality of state paths across participants.

**Between subject differences in dynamics link to movie ratings.** We then investigated if brain state dynamics unique to each participant were associated with their subjective ratings of the movie. Subjective ratings were obtained using a simple questionnaire containing questions about (i) boredom, (ii) enjoyment, (iii) emotional feelings, and (iv) audio quality. All questions could be answered using a 1 to 5 rating scale. To investigate the link between movie experience and brain state dynamics, we used an inter-subject representational similarity analysis (IS-RSA[33,34], see Methods) which looks at the representation across participants with an inter-subject distance matrix. The IS-RSA tests whether participants who reported similar subjective experiences also had comparable brain dynamics.

We first characterised the questionnaire responses using a multidimensional scaling approach (Methods). In brief, this method visualises the dissimilarity between individual movie ratings as distances between points in a two-dimensional plane. This analyses showed that movie experience was variable across participants, ranging from high engagement (low boredom, high

states 5 and 9 and lesser occupancy of brain states 1–4 and 6–8. The inter-subject consistency of the brain state expression was also lower (Fig. 2 and Supplementary Fig. 6), with each participant displaying a unique brain state progression through time during the rest condition. In addition, qualitatively, brain states 5 and 9 were visited for a longer time (Fig. 4b). Contrary to resting state, movie-induced brain state dynamics showed an

**Table 3 Brain state profiles, determined by their overlap with story annotations.**

|  | Faces positive | Faces negative | Scenes positive | Scenes negative | Language | Changepoint |
|---|---|---|---|---|---|---|
| State 1 | 13.17 |  | 14.22 |  |  |  |
| State 2 |  | 13.08 | 7.09 | 11.40 |  | 5.58 |
| State 3 |  | 24.19 | 11.96 | 11.31 | 39.42 |  |
| State 4 |  |  |  |  |  |  |
| State 5 |  |  |  |  |  |  |
| State 6 |  | 9.31 |  | 12.58 |  |  |
| State 7 |  |  |  |  |  | 9.70 |

The $t$-statistics of the overlap between brain states and movie annotations are significant at $p_{FWE} < 0.05$ (equivalent to uncorrected $p < 0.00083$ and $t > 3.16$).
States 8 to 10 exhibited no significant associations with movie annotations.

**Fig. 4 Dynamic characteristics of brain states across rest and movie viewing for session A.** Session B is presented in Supplementary Fig. 3. Asterisks indicate statistical significance (paired two-sided $t$-tests, $p_{FWE} < 0.05$; corrected for multiple comparison across 10 states; $n = 14$ participants examined over two consecutive sessions: rest session A and movie session A). **a** Fractional occupancy. States 1–4 and 6–8 have higher occupancy in movie viewing, while the opposite is true for states 5 and 9. The exact $p$ values for states S1–S10 are: $1.8e^{-7}$; $7.0e^{-7}$; $1.6e^{-4}$; $8.7e^{-4}$; $3.7e^{-3}$; $7.2e^{-5}$; $4.7e^{-5}$; $3.3e^{-3}$; 0.13. **b** State dwell times. Qualitatively, the dwell time for states 5 and 9 are higher in rest condition. Grey lines show how occupancy and dwell times are paired within participants. Boxplots: upper (lower) box edge: 25th (75th) percentile; central line: median; dotted lines: 1.5× interquartile length; whiskers extend to the most extreme data points not considered outliers; red plus: outliers.

enjoyment, high emotion, and high audio quality) to low engagement (Fig. 6a). In IS-RSA, movie engagement was represented with the full subject-by-subject distance matrix of questionnaire answers (Fig. 6b).

We then applied RSA to compare the representation of individual differences in brain state dynamics (FO and state transitions) with the representation of the individual ratings of the movie (Fig. 7a, "Methods"). Differences in FO and questionnaire representation were positively correlated ($r = 0.174$, $p = 0.031$). We also observed a positive correlation between the between-subject distance in the pattern of brain state transitions and answers to the post-movie questionnaire ($r = 0.182$, $p = 0.034$) (Fig. 7b). There was no statistically significant association between the state path dissimilarity and the movie ratings ($r = -0.09$, $p = 0.27$). Whereas the questionnaire representation can be divided up into more or less engagement in the movie (Fig. 6), the brain dynamics representation can be divided

into increased expressions of brain state 1, 7, and 8 combined with reduced expressions of brain states 2 and 3, and vice versa. This finding is associated with more transitions from state 3 to state 4. These results appeared to be specific to movie viewing as brain state dynamics extracted from the pre-movie resting-state data did not significantly correlate with movie appraisal.

## Discussion

The brain is constantly active, with activity across disparate brain regions supporting diverse cognitive functions, which in turn allow appraisal and interaction with the environment[6,35–37]. Here, we assessed if the temporal dynamics of distinct brain states reflect the sensory, cognitive, emotional, and physiological processes underpinning the subjective experience of a theatrical movie. In particular, we contrasted resting state and movie-viewing acquisitions to understand how perceptual immersion in

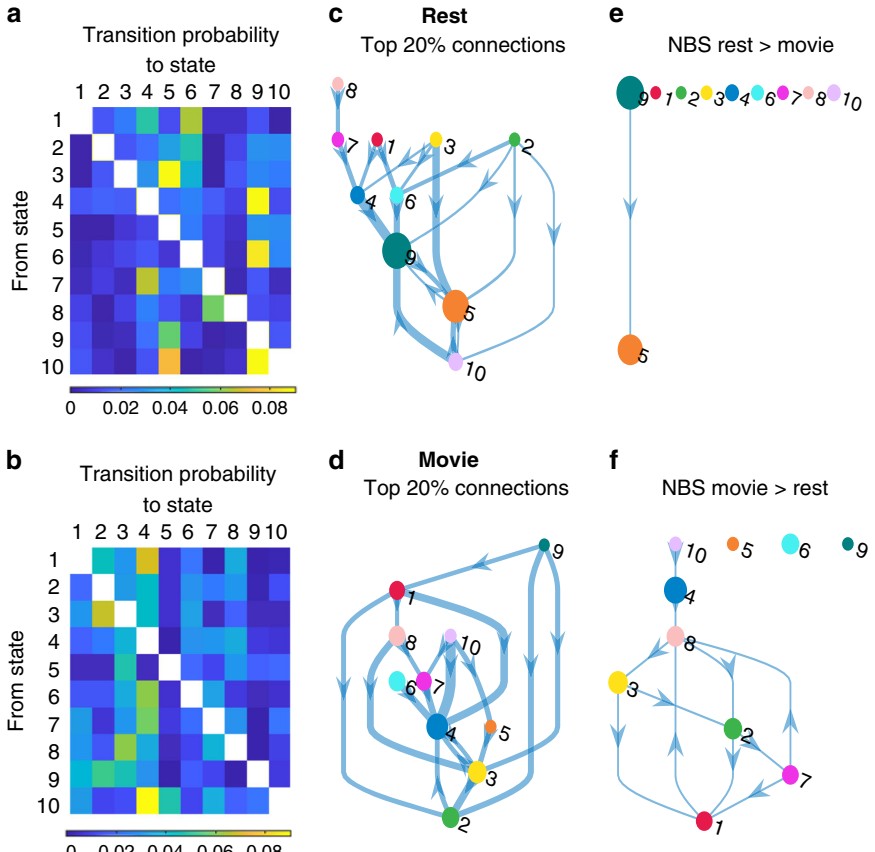

**Fig. 5 Brain state transition probabilities during rest and movie viewing.** Left column: Group averaged transition probability matrices for rest (**a**) and movie viewing (**b**). Diagonal elements are omitted for clarity. Middle column: Top 20% state transitions during rest (**c**) and movie viewing (**d**). Arrow thickness corresponds to the group-averaged probability of that transition. Right column: State transitions with a significantly higher probability of occurring during rest than during movie viewing (**e**) and vice versa (**f**) as identified by the Network-Based Statistics ($p_{FWE} < 0.05$). Each of the colour circles represents a state according to the colour scheme used in Fig. 1. The diameter is scaled according to that state's fractional occupancy (Fig. 4a). Each arrow represents a transition.

an engaging naturalistic stimulus reshapes whole-brain state transitions. Using a meta-analytic approach, we first found that movie-viewing elicited a richer repertoire of HMM brain states than at rest, which matched distinct functional profiles. The temporal expression of these profiles recapitulated the content of movie scenes, was associated with distinct physiological changes and correlated with the subjective appraisal of the movie. These results demonstrated that characterisation of brain state transitions can inform our understanding of exteroceptive and interoceptive processes that support the appraisal of an ecologically valid sensory experience.

Movie viewing allows the study of transitory brain states linked to an immersive sensory stimulation[3,38]. This experimental paradigm provides a unique opportunity to capture complex brain dynamics that may not otherwise be detectable through the lens of traditional task designs[18]. Due to the complexity of capturing meaningful brain states and temporal dynamics, neuroimaging research has thus far largely adopted resting-state paradigms[6,39,40]. Using a number of complementary techniques, these recent studies have suggested the existence of discrete brain states that are characterised by the differential expression of patterns of brain activity in regions comprising canonical brain networks (BNs)[3,6,9,41–43]. However, the functional relevance of these spontaneous brain states and their context-driven temporal expression has not yet been resolved. To address this, we adopted an approach that allows the characterisation of whole-brain transitions between inferred brain states[5]. This method has recently been used to describe slow (seconds[44]) and fast

(milliseconds[39,45]) resting-state dynamics that reflect genetic and behavioural traits, including intelligence[5]. The current work demonstrates that this generative modelling technique can capture subject-specific brain state transitions that are temporally aligned to perceptual, semantic, and narrative features of a movie.

The expression of brain states is thought to support defined functional network profiles required for the current sensory context. Hence, such brain states should reliably emerge during similar sensory experiences. We tested these hypotheses by adopting a multi-session protocol involving repeated resting state and movie viewing tasks, each interleaved by three months. Our findings firstly support recent observations that resting-state brain dynamics are predominantly bistable, converging with prior models of resting-state EEG[46] and fMRI data[5]. Specifically, we found that two dominant states at rest whose network expressions reflected only subtle modulations of the mean network activities across all acquisitions. These findings are broadly compatible with results showing structured fluctuations in resting-state fMRI data[47], while also highlighting that such dynamics are significantly less rich and structured than those induced by complex audiovisual stimuli. Indeed, movie viewing imposed richer brain state dynamics that were characterised by distinct functional profiles, with stronger perturbation away from the global mean activity, which is consistent with deeper attractor networks[48]. These state transitions were temporally aligned with the narrative structure of the movie, displayed a close association with corresponding sensory, perceptual, and emotional content, and mirrored changes in HR variability and PD. The link between

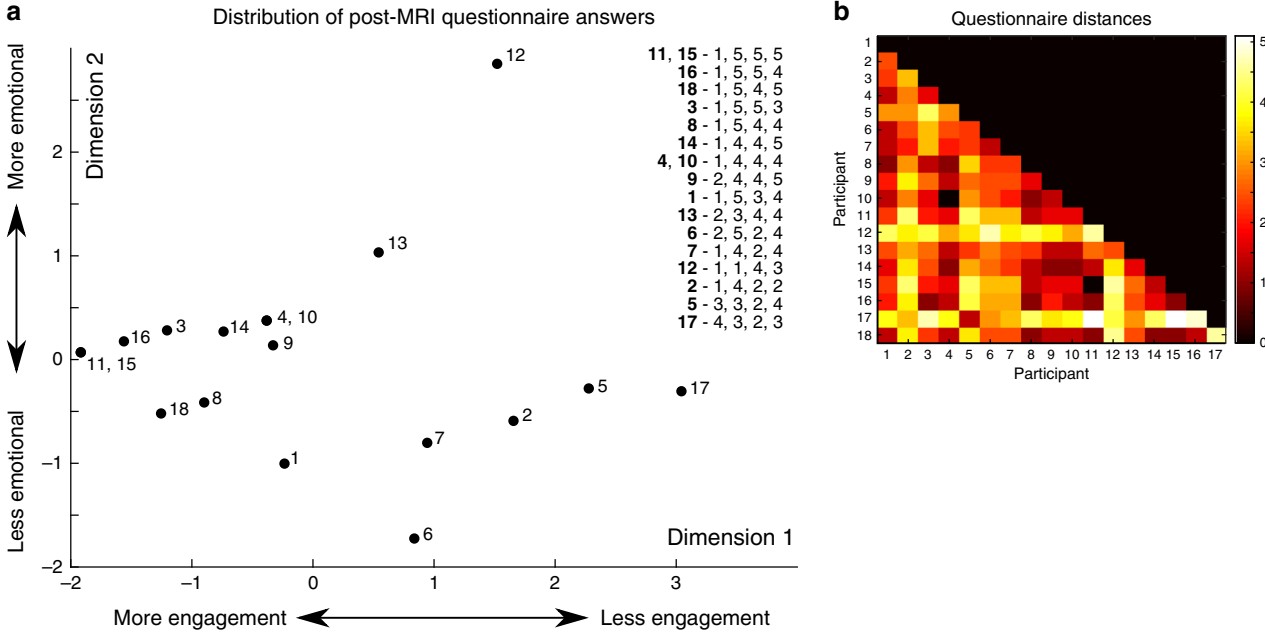

**Fig. 6 Analysis of the questionnaire answers using multidimensional scaling. a** The inter-subject distances of the answers are plotted as points on a 2D plane. Numbers denote participants. The answers are given in the inset and coded for (in order): Participant number—boredom (1–5), enjoyment (1–5), evoking emotion (1–5) and audio quality (1–5). Answers can be scaled from left to right as participants who are more engaged in the movie answer lower on boredom and higher on enjoyment, emotion and audio quality. Scaling from top to bottom is based on how participants scored for induced emotion. **b** Inter-subject distance matrix of the post-movie questionnaire answers.

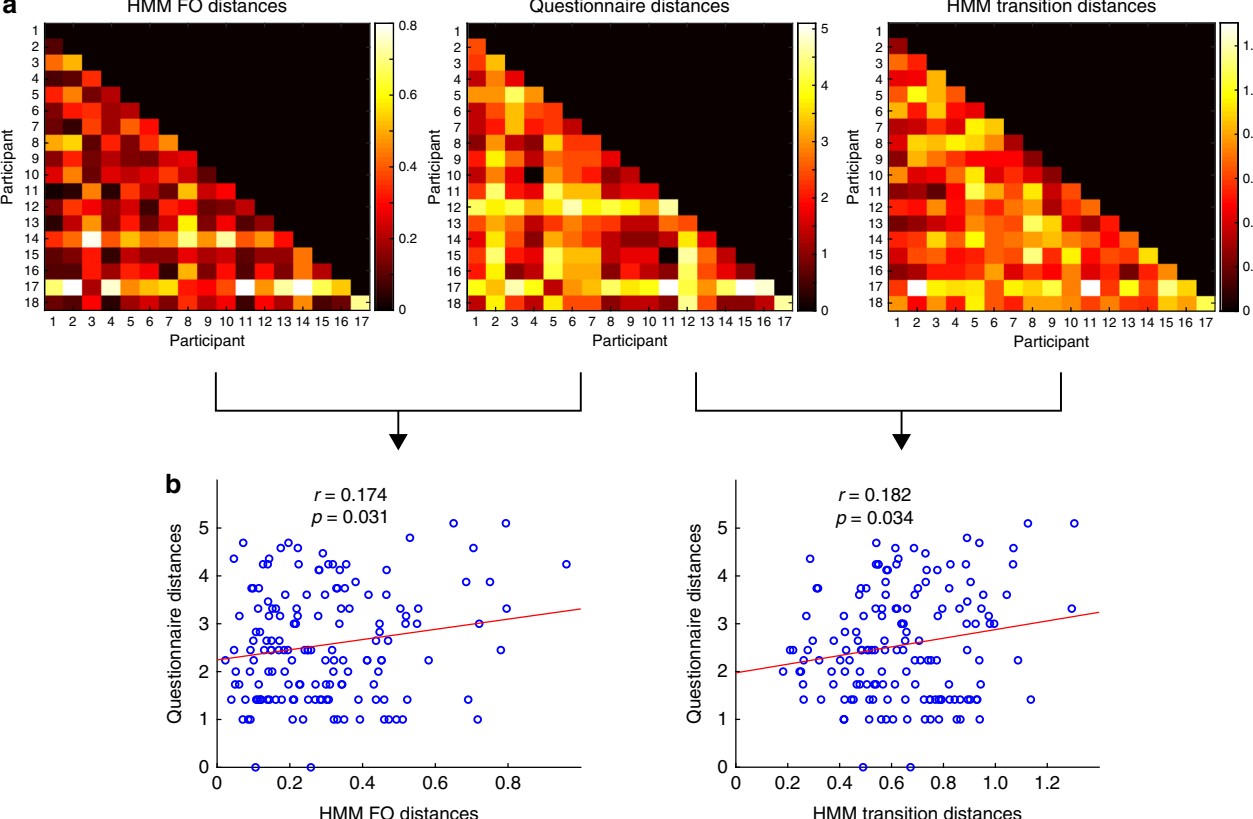

**Fig. 7 Association between brain state dynamics and behaviour. a** Representations across the participant sample of fractional occupancy (FO), questionnaire responses, and state transition matrices. These are given by the inter-subject distance matrix, where each element is a pair-wise distance. Correlation distance is used for FO and state transitions, whereas the Euclidean distance was used for questionnaire distances. **b** Pearson's correlation between the representations of questionnaire answers and FO (left) and state transitions (right).

physiological changes and brain state dynamics suggest that the sensory properties of a movie, as well as the content of its narrative, could be manipulated to evoke discrete brain processes. These findings motivate further investigations into the use of structured naturalistic stimuli to induce sequences of brain states underpinning a broad range of sensory and cognitive processes.

Notably, our data showed that subject-specific idiosyncrasies in brain states FO and state transitions correlated with the subjective rating of the movie. In particular, the correlation only involved brain states that were more expressed during movie viewing compared to rest. This result adds weight to the notion that HMM brain states and dynamics reflect subjective sensory and perceptual computations and their integration with higher-order cognitive and emotional processes. This finding also highlights that discrete snapshots of patterns of brain activity, isolated using relatively simple low dimensional representations, can capture the complex brain dynamics underpinning our rich subjective experience. Previous work has shown that the recall of a movie's content is associated with inter-subject correlation in fMRI time series during movie viewing[49]. Our work adds to this by showing that recall of emotional responses are also linked to common brain state transitions.

We sought to assess if structured departures from patterns of spontaneous brain activity that emerge when viewing an ecologically valid sensory experience are functionally meaningful and reproducible. To this end, we focused on 14 canonical BNs because such large-scale systems have been consistently implicated at rest and during cognitive, emotional, motor, and perceptual tasks[50–52]. These prior findings suggest that these canonical networks capture core properties of functional brain organisation[53,54]. This spatial scale of description for whole-brain dynamics has become ubiquitous in the field of cognitive and clinical sciences, facilitating the link between our findings and the existing literature and its future translation to clinical investigations. Other approaches have started from higher dimensional representations[43] or have chosen a data-driven, adaptive dimension reduction[55]. Although we identified rich multi-state dynamics during movie viewing, it is likely that the coarse-grained dimension reduction to 14 networks contributed to the restriction of resting-state dynamics largely to two states, less than previously observed[43,56]. Similarly, defining HMM states across large networks and estimating the model from whole-brain data imposes coordinated, whole-brain transitions. This precludes the identification of hierarchically nested time scales previously reported in movie viewing fMRI when HMM models were estimated regionally and not globally[19,57]. In sum, our choice of the spatial aperture of large functional networks is well tuned to highlight the transition from resting-state dynamics to those evoked by movie immersion. Alternative approaches reveal other complex features of these rich neuronal dynamics, including the role of state transitions in scene completion[58,59], narrative segmentation[60], and memory encoding[19].

Several caveats need to be considered when interpreting our findings. The HMM is a statistical technique that is not grounded on biophysical models of neural activity. The detected dynamics are the result of a statistical fitting of the data that imposes a strict assumption of discretely expressed, not continually mixed, brain states. In the resting state, fMRI dynamics have been previously described using twelve distinct brain states[5]. Our selection of ten states was based on control tests assessing the minimum number of brain states necessary to describe the data without redundancy. This was formally done by fitting the model and computing the Akaike Information Criterion, as well as assessing changes in the free energy under permutation-based testing[56] (see Methods). Moreover, we performed the HMM inversions numerous times, with each inversion revealing similarly structured brain state

dynamics across both participants and sessions. Due to the complex multi-session and multi-modal nature of the design, the current study comprises a relatively small sample size. Although our analyses were replicated across two temporally distinct recordings in the same participants, it is likely that more nuanced differences in brain states and their links to physiology and behaviour would be evident in a larger data set. Finally, the vexed issue of physiological confounds on functional BNs needs to be acknowledged[61]. Notably, these effects have most often been identified in resting-state acquisitions which, by virtue of their less constrained nature, challenge the disambiguation of nuisance effects from those due to visceral efferents such as the autonomic correlates of suspense, fear, or surprise[62,63]. Movie viewing may mitigate some of this concern because it comprises structured, emotionally salient material, which engenders physiological effects that are correlated with activity in central visceral centres such as the anterior insula[18].

Our findings recapitulate the need to consider distinct, non-stationary, patterns of brain activity to characterise the neural underpinnings of complex perceptual processes. Clinical translation of neuroimaging protocols place a heavy emphasis on test–retest reliability to make sure that variance between measurements can be reliably attributed to a disease state or progression[64]. Contrary to the commonly used resting-state paradigm, we demonstrate that brain state dynamics isolated using a movie paradigm have markedly higher test–retest reliability. This finding encourages the uptake of ecological paradigms and time-resolved methods by clinical studies interested in assessing changes in whole-brain brain dynamics as a function of disease state and progression.

## Methods

**Participants.** Twenty-one healthy participants (11 females, right-handed, 21–31 years, mean age 27 ± 2.7 years) were recruited for the study's two movie viewing sessions. 17 participants completed both movie viewing sessions. Three participants were excluded because of in-scanner head motion.

The study was approved by the human ethics research committee of the University of Queensland and written consent was obtained for all participants.

**Experimental paradigm.** The experiment comprised two scanning sessions, three months apart. For each session, fMRI data were acquired from participants during an 8-min eyes closed resting state session, followed by viewing of a 20-min short movie called the The Butterfly Circus using the Presentation 16.3 software. Details regarding the experimental design have been reported elsewhere[42]. In short, the Butterfly Circus narrates an intense, emotionally evocative story of a man born without limbs who is encouraged by the showman of a renowned circus to overcome obstacles of self-worth and reach his own potential. The narrative architecture of The Butterfly Circus map onto three distinctive drama acts with significant developments, complications, and turning points for each act[42] (Supplementary Table 1). Moreover, the following basic annotations were provided with the data: (i) the use of language, (ii) change of scenes, (iii–v) Positive/Negative Faces, and (vi–viii) Positive/Negative Scenes (details in Supplementary Fig. 11).

**Image and electrophysiological acquisition.** Data were acquired using a Siemens TIM Trio scanner equipped with a 12-channel head coil[42]. The gradient-echo echo planar-imaging (EPI) scanning sequence had a repetition time (TR) of 2200 msec and a resolution of 3 mm$^3$. For the rest condition, 220 EPI brain volumes were acquired, whereas, for the movie viewing, 535 volumes were acquired. A T1-weighted structural image covering the entire brain was also collected (resolution of 1 mm$^3$).

Concurrent with functional imaging, (electro-)physiological recordings were also acquired: (i) HR was obtained from a Brain Products MR-compatible BrainAmp amplifier (Brain Products GmbH, Gilching, Germany) with a sampling frequency of 5000 Hz; (ii) respiration was obtained from the Scanner's Personal Monitoring Units system with a sampling frequency of 50 Hz; and (iii) PD was recorded using an MR-compatible Eyelink eyetracker (EyeLink SR Research) with a sampling frequency of 1000 Hz. Due to the recording quality, only the HR and PD were used to link brain state dynamics with physiological changes.

**Image and electrophysiological data processing.** Image preprocessing was performed using fMRIPrep version 1.1.5[65], which is based on the Python toolbox Nipype. Structural images were first corrected for intensity non-uniformity and

spatially normalised to MNI space (ICBM 152 Nonlinear Asymmetrical template version 2009c). Brain tissue segmentation of cerebrospinal fluid (CSF), white matter (WM), and grey-matter (GM) was also performed. Functional images were slice-time corrected, motion corrected, co-registered to the structural image, normalised to MNI space, and spatially smoothed with a 6 mm Gaussian kernel. ICA-AROMA was subsequently performed using non-aggressive denoising[66]. Two confounding time-series were obtained from functional image preprocessing: global signal in WM, and global signal in the CSF. After spatial preprocessing, temporal preprocessing was performed with the toolbox Nilearn version 0.5.0. Temporal preprocessing included filtering the data between 0.01 and 0.15 Hz to capture the neural signal associated to both rest and task[67] and regression of global WM and CSF signals.

HR was preprocessed using FMRIB FastR[68] to detect heart-beat events. The Tapas IO Toolbox version 2016[69] was used to measure HR at the same temporal resolution as the fMRI time series (2200 ms). PD preprocessing was done using custom MATLAB scripts (available here: [https://github.com/brain-modelling-group/MovieBrainDynamics]) to detect eyeblinks and bad data segments. These data were then interpolated and downsampled from 100 Hz to the same time resolution as the fMRI time series.

**Hidden Markov Model (HMM).** The basic premise of the HMM is that dynamic fluctuations in BOLD-inferred neural activity within brain regions comprising the 14 canonical BNs can be decomposed into a sequence of discrete hidden brain states that switch and recur over time according to a time-invariant transition probability. Therefore, each time point can be classified as belonging to a single brain state, which represents a whole-brain configuration of average fMRI signal across the 14 BNs.

For each of the 14 BNs[23], time series were obtained by: (i) calculating the mean signal within the ROIs comprising the network; (ii) removing time series associated to the first five volumes; (iii) demeaning the signal; and (iv) scaling the resulting time series by its standard deviation. Thus, network-averaged fMRI time series of zero mean and unit standard deviation were calculated for each participant and session (two movies and two resting-state conditions). Next, each participant's time series were temporally concatenated to a time × BNs matrix (14 participants, each with 215 volumes (8 min) of rest A, 530 volumes (20 min) of first movie viewing, 215 volumes (8 min) of rest B, and 530 volumes (20 min) of second movie viewing: 20,860 × 14 matrix). The HMM was then fitted to the temporally concatenated time courses (allowing for covariance between the time series) to yield a single set of 10 model parameters (brain states). This concatenation strategy allows participants to have the same set of brain states across movies and rest periods, maintaining the ability to assess brain state dynamics across conditions. These are assessed in terms of the: (i) total time spent in a state over a longer time period (FO); (ii) time spent by each participant in a given state before switching (dwell time); and (iii) likelihood of switching between specific states (transition probability).

The HMM-MAR MATLAB toolbox ([https://github.com/OHBA-analysis/HMM-MAR]; commit version 7a5915c) was used to perform Variational Bayes inversion on the HMM using 500 training cycles, according to previously established procedures[5,43,56]. The HMM assumes that fMRI time series can be described using a dynamic sequence of a limited number of brain states[5]. The total number of states needs to be specified a priori. Previous studies modelling fMRI dynamics in healthy individuals considered between 5 and 12 states[5,54,56]. In our analysis the HMM input was a 20,860 (14 participants with every 1490 timepoints) by 14 (average signal from 14 network masks) matrix. In this particular instantiation of the HMM, each brain state was defined by a multivariate Gaussian distribution, which was described by the mean within each voxel, and covariance between voxels, when each state is active[5]. Similar to the procedure found in previous work[56], we used the Akaike Information Criterion (AIC) metric to infer the HMM with 10 states. In addition, we also inferred the HMM with 6, 8, 12, 14 and 25 states, with each state choice decoded 15 times. We found that using 12 states or more yielded HMM results in which several states were not occupied. Therefore, the use of 12 or more likely reached a practical limit on what the HMM could decode in our current dataset.

HMM inversion yields three different types of output; i.e., structural, temporal, and dynamic. The structural output is a heat map for each brain state with the fMRI signal across the 14 BNs (Fig. 1), together with the functional connectivity/covariance matrix. The variability in the fMRI signal was used to estimate distinct patterns of BN activity (brain states). The temporal output is the state path (Fig. 2) for each participant and session[56]. This output provides information on the most expressed brain state at each time point during rest and movie watching. The dynamic output is generated from the state paths and parameterises the dynamics across the rest or movie periods with: (i) fractional occupancy and state dwell-times per brain state (Fig. 4) and (ii) state transition matrices (one for each participant and session) encoding for the probability of transitioning from one state to another (Fig. 5, note that the matrix is not symmetric).

**Spatial definition of the 14 canonical BNs.** The HMM was used to model the temporal dynamics of 14 canonical BNs during the first and second movie viewing sessions, as well as the rest condition. The spatial extent of these BNs were defined according to an established reference definition that maximally disambiguates cognitive states[23] (Supplementary Fig. 1). For each participant and BN, the fMRI

signal was averaged across all voxels to construct an N (time points)-by- M (14 BNs) matrix containing the time courses of each BN. These time courses were then normalised to have zero mean and unit standard deviation across time. Normalisation was done separately for each BN and participant. Thus, per participant, we obtained two matrices (535 × 14) for movie viewing and two matrices (220 × 14) for the resting state.

**Reverse Inference of brain states with Neurosynth decoding.** The Neurosynth database contains nearly 14,300 fMRI studies and 507,000 reports of BOLD-inferred brain activity ([www.neurosynth.org]). By mapping keywords (topics) extracted from the literature body to the locations of the brain activity, it enables broad cognitive functions/states to be decoded from brain activity in entire studies or individual participants[22]. The decoding itself is an association test such that it reflects the probability of a psychological process being present given the pattern of activity over several regions in the brain. The Neurosynth framework provides a comprehensive set of whole-brain term-to-activation 'topic' maps that allow one to calculate either forward associations (P(Activation|State)) or reverse associations (P(State|Activation)). In the present study, we forward associated the brain state of our HMM model (Fig. 1) to the topic maps of 16 general terms chosen to encompass a variety of brain processes applicable to movie viewing[24].

We correlated the spatial distribution of each brain state (Fig. 1) to the topic maps, effectively de-coding the range of mental states associated with each brain state during watching of the Butterfly Circus movie (Fig. 3). Decoding was performed using a python notebook obtained from the Neurosynth's Github webpage ([https://github.com/neurosynth/neurosynth]; commit version 948ce7).

**Consistency of state paths over participants and sessions.** In order to calculate the consistency between participants of each brain state's path over time (Fig. 2), we used a sliding window of nine consecutive BOLD volumes (19.8 s). Within this window, for each of the 10 brain states, we counted the number of participants that had this state expressed at least once and identified the most frequently expressed state. This information is displayed in the bottom trace of Fig. 2. We then contrasted the states expressed during movie watching with those of the resting-state scans. Even though there were fewer states expressed during resting state, the consistency count was markedly lower (only reaching 50–65%, i.e., 7–9 out of 14 participants, at certain times). However, during movie viewing, the count reached 100% multiple times (Table 1).

In order to calculate the consistency of brain states over sessions, for each state and each participant, we constructed a binary vector of 0 (brain state not expressed) and 1 (brain state is expressed) for each time-point during movie viewing (530 scans) and rest (215 scans) for both session A and session B. To assess the consistency over sessions (A and B), we calculated the Jaccard Overlap index between the binary vectors for each brain state and averaged them. A paired t-test was performed on these values to test whether Movie viewing has more consistent brain-state overlaps than resting state.

**Comparison of state dynamics between rest and movie viewing.** In order to assess brain state dynamics between rest and movie viewing, we compared the dynamic metrics (FO and state dwell time) for each brain state. Significant differences between these conditions were assessed with paired t-tests and marked in the figures with an asterisk. We also compared the state transition probabilities (averaged over participants) between the conditions in Fig. 5. A threshold of 20% was applied to identify the most frequent transitions (middle column, Fig. 5c, d). Finally, the Network-Based Statistics toolbox (version 1.2) was used to reveal the network of state transitions that are significantly more expressed during movie watching compared to rest, and conversely during resting state compared to movie viewing (right column).

**Link between brain states and electrophysiological data.** In order to examine whether HR and PD were consistently increased or decreased during brain state visits, the following procedure was used. The HR and PD values were segmented according to the brain state path, and averaged across time to calculate (for each brain state) the value during a visit to brain states 1–10. Then, across the entire movie averages were calculated for HR and PD to obtain a baseline, which was subtracted from the brain state-specific value to generate the deviation value for heart rate (ΔHR) and pupil diameter (ΔPD); see Table 2. This produced 14 deviation values for each brain state (one for each participant). A one-sample two-sided t-test was then used to assess the likelihood of the observed deviations against zero deviation.

**Association between brain states and movie annotations.** To obtain information about the putative connection between HMM brain states and movie annotations reflecting the unfolding narrative, we calculated the Szymkiewicz–Simpson overlap[70] between two state vectors as shown in Supplementary Fig. 12. The HMM brain state vector composed of values 0 and 1 when the HMM brain state is not expressed or expressed, respectively (Supplementary Fig. 12b). The story annotations were also converted to vectors of 0 and 1 according to their onset and offset times (Supplementary Fig. 11). Annotations vectors were generated for Positive

Faces, Negative Faces, Positive Scenes, Negative Scenes, Language, and Changepoint.

In order to calculate the significance, the averaged overlap index across participants was compared to a null distribution of overlap indices generated by 5000 permutations of the brain state vectors and their overlaps with the annotation vector. For each iteration, the movie annotation vector was randomly shuffled, the overlap between the HMM state vector and annotation vector was re-calculated, and finally averaged over participants to create a new random value. The permutations yielded a null distribution that allowed for a comparison with the observed overlap index to infer statistical significance via a *t*-score (Supplementary Fig. 12c). All significant findings are reported in Table 3.

**Post-movie questionnaire**. The post-movie questionnaire had eight questions. Four questions assessed the following: whether the participants had seen the movie before (none had watched it before), if English was their native language (yes for all participants), how participants rated their English fluency (two participants indicated their fluency as 4 out of 5, whereas the remaining participants indicated 5 out of 5), and how well the participants understood the movie content (all indicated an understanding of 100%). These questions were omitted for the RSA analysis due to the lack of variability across participants. The four remaining questions more closely assessed the subjective appraisal of the movie and were used for the RSA:

1. Did you get bored during the 1st movie session?
   Not at all: 1 2 3 4 5: very bored
2. How well did you enjoy the 1st movie session?
   Not at all: 1 2 3 4 5: very enjoy
3. How emotional did you feel during the 1st movie?
   Very sad: 1 2 3 4 5: very happy
4. How was the audio quality for the 1st movie?
   Very poor: 1 2 3 4 5: very good

Pertaining to question 4, the audio quality itself did not change during recordings and all participants reported that they understood the movie content in a similar fashion. Participants likely rated this scale differently depending on their own engagement. To quantify participant differences in movie evaluation, a multidimensional scaling approach was applied to plot the questionnaire answers as points in a two-dimensional representation (engagement by evoked emotion). This approach revealed that there was a group of participants who were more engaged with the movie (i.e., low boredom score combined with high enjoyment, emotional and audio quality scores) and another group who were less engaged (i.e., having high boredom or a low score on the other three questions); see Fig. 6.

**Association between movie ratings and brain state dynamics**. We used the inter-subject representational similarity analysis (IS-RSA) to assess how the brain and behavioural data are represented in a group sample. Four inter-subject distance matrices (each $18 \times 17$) were constructed for representations of brain dynamics (FO and state transition), brain state Viterbi path, and movie impressions (Fig. 7). To calculate the inter-subject distances for movie impression (session A), the Euclidean distance of questionnaire ratings between each possible pair of participants was measured producing the 18 (participants) × 17 matrix. For the FO representation, for every possible pair of participants, the correlation between the 10 FO values (one for each state) was calculated to produce the inter-subject distance matrix. For the state transition representation, for every possible pair of participants, the correlation between the state transition matrices (10 states × 9 transitions to another state) was calculated to produce the inter-subject distance matrix. Finally, for the brain state (Viterbi) path, the Jaccard dissimilarity index (1-Jaccard index) was used. In this way we generated a single representation for movie impressions, two representations for brain state dynamics (one for FO and one for state transitions) and one representation for brain state paths. In order to assess the strength of associations between the brain states and movie rating representations, we calculate the Pearson's correlation between the lower triangular parts of the matrices shown in Fig. 7a. Statistical significance was assessed using permutations (i.e., re-calculating the correlation 5000 times, with each permutation shuffling the participants in the movie rating matrix). From the null distribution, Z score (and associated *p*-values) were obtained.

**Reporting summary**. Further information on research design is available in the Nature Research Reporting Summary linked to this article.

## Data availability
Extracted fMRI time-series used to infer the HMM, regions of interest, extracted physiological parameters, questionnaire answers and movie annotations are available in the "data" folder of the following Github repository: [https://github.com/brain-modelling-group/MovieBrainDynamics]. The Github repository has the following DOI: [https://zenodo.org/badge/latestdoi/244780030]. The repository contains all preprocessed fMRI and physiological data needed to reproduce the results. The original functional and structural MR images will be made available upon reasonable request to the authors with mandatory ethics approval and data sharing agreement with QIMR Berghofer.

The web interface to the Neurosynth database is available here: [www.neurosynth.org]. The Neurosynth database is also accessible using the python interface mentioned in the Code availability section.

A reporting summary for this Article is available as a Supplementary Information file. Source data are provided with this paper.

## Code availability
The code used to run the HMM inversions and generate Figs. 1–7 is provided in the following Github repository: [https://github.com/brain-modelling-group/MovieBrainDynamics]. This repository contains forks of (a) the Github repository containing the Matlab code used to perform HMM inversions located at [https://github.com/OHBA-analysis/HMM-MAR] and (b) the Github repository containing the python code used to query the Neurosynth database, located at [https://github.com/neurosynth/neurosynth].

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

## Acknowledgements

The authors are particularly grateful to Christine C. Guo and Vinh T. Nguyen for providing the data. L.C. is supported by the Australian National Health Medical Research Council (L.C. 1099082 and 1138711). J.N.v.d.M is supported by the Australian Research Council Centre of Excellence in Integrative Brain Function (CIBF) CE140100007. L.J.C. is supported by the National Institute of Health (R01MH116026). M.B. is supported by an NHMRC Fellowship (1118153).

## Author contributions

J.N.v.d.M., M.B. and L.C. designed the research; J.N.v.d.M. performed the analysis; L.J.C. and L.C. contributed analysis approaches; S.S. organised data collection; all authors wrote the paper.

## Competing interests

The authors declare no competing interests.
