## [Peer Review File · Nature Communications]

Reviewers' Comments:

Reviewer #1:

Remarks to the Author:

In this paper, the authors present a study on the use of brain state dynamics in the context of movie data, which is argued to be a compromise between completely unconstrained cognition paradigm (resting-state) and non-ecological classic task designs. The authors use Hidden Markov Modelling for the analysis, which they ground on theoretical ideas of metastability and multistability.

The study is well executed, the use of the methods is sound, and the authors have put care on validation, for example through test-retest reliability analyses. The relation to autonomic indices and subjective ratings is also interesting. I have a few suggestions that I hope would be useful.

In my opinion, given that this kind of modelling has already been performed in movie data (see Baldasano, Chen, et al, 2017; Neuron 95 - which the authors should probably reference) and of course on resting state (Vidaurre et al. 2017 - which the authors already reference), the real novelty of this paper is in having a thorough and solid understanding on the differences between movie-watching and resting-state.

For example, it seems that during resting there are fewer states, and less switching. What's the reason for that? Is this a biological reason, e.g. because the brain is "more relaxed" somehow in rest? or is it because the HMM has a harder time in describing the data in the resting-state, due to the data having less structure? In the latter case, the inference of the model would probably put more explanatory power on the movie data, and sort of "give up" on the resting-state. One way to investigate this is to compute the likelihood of the model per time point, and see how it differs between rest and movie data (this is the second output argument of the function `hmmfe`, which is in the toolbox the authors are using). A higher likelihood would mean "better explained".

Another analysis that would help in this regard is to run the HMM separately on rest and movie, and compare summary statistics such as the amount of state switching.

An important confound that the authors are seemingly not considering is the different amount of data between the two conditions, given that there are more movie data (20min) than resting state data (8min). Do the results hold when only 8min of data are used in the inference? If brain activity is different between these two conditions (which is the case), having much more data for one of the conditions would trivially lead to solutions where the condition with less data uses fewer states.

Also, I would find very interesting to compare inter-session consistency (for each subject separately and averaged across subjects) versus the inter-subject consistency (for each session separately and averaged across sessions). I wonder which one is higher. If the inter-session consistency is higher, that might suggest that there's a movie-related subject-specific neural signature, which I think is intriguing.

Other suggestions:

- pp13, line260. "In addition, these visited states were characterized by strong and specific functional network weightings." Do the authors mean functional connectivity? Are differences in FC reported? Sorry if I've missed this. I think it would be very relevant if that were the case, given the controversy about whether there are meaningful differences in FC or not.

- A technical point: could the authors specify upfront the observation model of the HMM? Is it a

Gaussian distribution with mean and covariance?

Diego Vidaurre

Reviewer #2:

Remarks to the Author:

In this paper the authors apply a Hidden Markov Modeling approach to characterize brain states during both resting-state and movie-watching data. This was an interesting paper to read, with some very nice and informative figures. There are a number of different analyses presented, which help characterize the discovered states in terms of their functional profiles, impacts on physiological measurements, and stimulus features, and compare dynamics between movies and resting-state as well as across subjects.

Major comments:

1) The framing of this paper is that work on understanding brain dynamics (on the scale of minutes) "has thus far largely adopted resting state paradigms," and that this work extends HMM approaches used for resting-state data to the movie domain. This is a strange framing for this work, since there has been a large amount of recent work characterizing brain dynamics in movies, many of which have also used HMMs (and some of which has even been conducted by some of this study's authors).

Examples of some of this work that was not discussed in this paper include:

-Applying HMMs to fMRI movie data: Baldassano et al. *Neuron* 2017; Baldassano et al. *JoN* 2018; Luke Chang et al. *bioRxiv* 2018 (cited in the paper, but only regarding the "reverse inference" meta-analysis); James Antony et al. *bioRxiv* 2020; Heusser, Fitzpatrick, Manning *bioRxiv* 2018

-Applying HMMs to other task data: Vidaurre et al. *Cerebral Cortex* 2018

-Another approach for identifying dynamics in movies: Tseng & Poppenk *bioRxiv* 2019

The authors do not need to cite all of these papers, but should better situate their work relative to this growing literature.

2) The HMM fits to the resting-state data are surprisingly poor, consisting primarily of states 5 and 9, which are both essentially constant activity across all ROIs. This is not consistent with the traditional literature on resting-state activity, which usually shows highly structured fluctuations e.g. those used to define the Default Mode Network. The authors claim that this bistability is similar to that described in Hansen et al. 2015, but the conclusions of that paper are quite different - that paper describes bistability in the *connectivity* matrix (not the activity patterns of individual timepoints) and shows substantial structure in which regions co-occur (e.g. see Fig 5 in Hansen et al.). One possibility for why the HMM fit is so poor in the resting-state data is because the HMM was fit to resting-state and movie data concatenated together, and the longer length of the movie data encouraged the model to preferentially identify states that were useful for the movie data only.

3) The representation of the brain state at each timepoint is downsampled to just 14 values, reflecting overall activity in large pre-defined networks. It is unclear why this was necessary, since the HMM could be applied to much higher-dimensional data at the level of individual brain parcels or even individual voxels (though a covariance type other than "full" would need to be selected as the "covtype" in the HMM-MAR toolbox for high-dimensional states). This low-dimensional representation of the brain will mask interesting pattern dynamics at the region level which we know carry information about brain states during movie-watching, e.g. Chen et al. *Nature Neuroscience* 2017.

4) The between-subject differences section uses only summary statistics from each subject's neural data, i.e. the overall frequency of occupying each state and the transition matrix between states.

However these measures ignore whether two subjects are actually occupying these states *at the same times* during the movie, which seems critical for assessing whether subjects are having a shared "movie experience" - e.g. just knowing that two subjects were both sad during 20% of the movie seems much less important than whether they thought the same scenes were sad. A measure related to the "consistency" calculation in Fig 2 would capture this kind of across-subject alignment.

Minor comments:

5) The text description of Fig 1 (lines 102-107) is very confusing, and doesn't seem to correspond to the figure. States are described as belonging to three clusters but it is not clear which states comprise each cluster, states 4 and 9 are described as "mean overall activity" but the figure describes only state 5 as mean activity (and state 9 as low activity), state 7 is described as low activity but the figure labels state 9 this way (since state 7 does have high visual activity).

6) One of the annotations compared to movie states is changepoints. However we might expect changepoints to correspond to *switches* between states rather than to a specific state itself - is it possible to assess whether changepoints are related to state switches (between any states or specific pairs of states)?

Reviewer #3:

Remarks to the Author:

Summary: The paper identified ten brain states from fMRI timeseries datasets acquired during 8 minutes of resting state scanning and 20 minutes of watching and listening to a movie. Heart rate, HRV, respiratory rate and volume eyetracking with pupillometry data were also acquired, along with post scan questionnaire ratings. 18 people were analyzed from a first session and 14 complete 2 sessions. Brain states were computed using a data-driven Hidden Markov Model framework that, as discussed, 'is not grounded in biophysical models of neural activity' and in which 'statistical fitting of the data that imposes a strict assumption of discretely expressed, not continually mixed, brain states'. The paper uses much reverse inference with respect to interpretation of brain states and highlights the value of movie viewing paradigms that could be extended e.g. to clinical studies as they provide 'rich and reliable' measures of brain dynamics.

Overall I judge that this paper makes an important and valuable contribution to neuroscience literature and understanding of human, building from (ICA) work of Bartels and Zeki 1999, and suggesting a practical (alternative to rsfMRI) approach to quantification and evaluation of brain dynamics measured using fMRI. The sequencing and dwell time of these brain states and the inferred relationship to distinct perceptual and cognitive processes provides novel insights

There are a number of points that I seek clarity about.

Typically when describing fMRI brain networks, people look for interregional correlations in fluctuating BOLD signal or concurrent activation across regions. I don't think these are necessarily the same thing. I therefore would value more guidance regarding how best to think about these brain states and the network activity - activity here is thus presumably the magnitude of the BOLD signal? viz: '...inversion of the HMM from these data yielded ten distinct states. To understand the functional expression of these states, we coded their respective loadings onto each of 14 widely studied canonical brain networks (see Supplementary Fig. 1). The expression of network activity was normalized so that zero corresponds to the average activity of that network across the movies and rest periods. The variability was scaled according to that network's standard deviation. Each network was normalized separately (Methods). What is the rationale for scaling against across the network?'

Again I would like more guidance as to how I should think about this neural terms, since different regions have quite different neural firing properties

It is not always immediately apparent where specific results (brain states) came from (though the figures are very useful) – e.g. HMM analysis of combined resting state and movie datasets over first session, (concatenated over both sessions of the 14 individuals, as indicated above.. why?) or independently in each session. The results text needs a little more explicit clarification. Related to that is the imbalance in time between the 8 minute resting state and 20 minute movie. Would the results change if the movie was only 8 minutes long? (particularly the comparative results, and regarding fractional occupancy).

The alignment of findings to physiological measures is slightly confusing: The abstract refers to heart rate variability, Table 2 refers to heart rate change not HRV, the discussion refers again to HRV. Both HR and HRV data were available along with respiration (frequency and depth). It is unclear on what grounds HR(V) and pupil diameter measures were subselected to related to the brain data: why not text all, or look for correspondence between multi-axis body state and brain data. Mention is made that heart rate and pupil diameter representing physiological indices of 'arousal'. This may be so in resting state (as might decreased HRV, changes in ventilation) but the issue is more complicated for dynamic audiovisuals aspects of movies. I think the link to physiology needs more consideration – not least as cerebrovascular dynamics have been argued to underpin emergence of some resting state brainwise networks.

A little more detail would be useful regarding the resting state (presumed eyes open + fixation cross) for pupillometry but I could not see this stated.

Reviewer #1:

In this paper, the authors present a study on the use of brain state dynamics in the context of movie data, which is argued to be a compromise between completely unconstrained cognition paradigm (resting-state) and non-ecological classic task designs. The authors use Hidden Markov Modelling for the analysis, which they ground on theoretical ideas of metastability and multistability.

The study is well executed, the use of the methods is sound, and the authors have put care on validation, for example through test-retest reliability analyses. The relation to autonomic indices and subjective ratings is also interesting. I have a few suggestions that I hope would be useful.

Authors (A): We appreciate the constructive appraisal of our work.

R1.1. In my opinion, given that this kind of modelling has already been performed in movie data (see Baldasano, Chen, et al, 2017; Neuron 95 - which the authors should probably reference) and of course on resting state (Vidaurre et al. 2017 - which the authors already reference), the real novelty of this paper is in having a thorough and solid understanding on the differences between movie-watching and resting-state.

A. We revised the introduction and to more clearly position the unique contribution of our work as elucidating the transition from rest to movie watching. We have also incorporated these prior studies of movie fMRI using the HMM into the Introduction, also noting the convergences and differences compared to the current approach:

Abstract

*Adaptive brain function requires that sensory impressions of the social and natural milieu be dynamically incorporated into intrinsic brain activity, where they are used to update internal models of the world and make predictions about the future. While dynamic switches between brain states have been well characterised in resting state acquisitions, the remodelling of these state transitions by engagement in naturalistic stimuli remains poorly understood. By exploiting the rich perceptual and emotional content of a short movie, we find that the temporal dynamics of brain states, as measured in fMRI, are reshaped from predominantly bistable transitions between two relatively indistinct states *at rest*, toward a sequence of well-defined functional states *during film viewing* whose transitions are temporally aligned to specific features of the movie. The *expression* of these brain states *covaries* with different physiological states (heart rate variability and pupil diameter) and reflects subjectively rated engagement in the film. In sum, a data-driven decoding of brain states reveals the distinct reshaping of functional network expression and reliable state *changes* that accompany *the transition from resting state to perceptual immersion* in an ecologically valid sensory experience. *Juxtaposing resting state with movie viewing acquisitions highlights the influence of naturalistic perceptual streams on brain-state transitions.**

Introduction

Page 4, paragraph 1:

Naturalistic stimuli, such as films⁴ and spoken narratives²⁴ offer the constraint and replicability that resting state acquisitions lack while adding greater ecological validity than traditional task designs^{25,26}. Recent analyses of movie viewing fMRI data using the HMM reveal a hierarchy of timescales, with more frequent state transitions in sensory cortex hierarchically nested within progressively slower transitions in heteromodal regions²⁷. Such multiscale dynamics mirror the statistics of the natural world²⁸ and suggest a remodelling of intrinsic correlations so that their complexity more closely matches the statistical structure of naturalistic perceptual streams^{5,29}.

Comparing unconstrained resting state acquisitions to film viewing holds potential to characterise this process, and hence to investigate the functional significance of transitory brain states.. [...]

Discussion

Page 18, paragraph 1:

The brain is constantly active, with activity across disparate brain regions supporting diverse cognitive functions, which in turn allow appraisal and interaction with the environment^{9,48-50}. Here, we assessed if the temporal dynamics of distinct brain states reflected the sensory, cognitive, emotional and physiological processes underpinning the subjective experience of a theatrical movie. In particular, we contrasted resting-state and movie-viewing acquisitions to understand how perceptual immersion in an engaging naturalistic stimulus reshapes whole-brain state transitions. Using a meta-analytic approach, we first found that movie-viewing elicits a richer repertoire of HMM brain states than those that occur at rest, and which match distinct functional profiles.

Page 18, paragraph 2:

However, the functional relevance of these spontaneous brain states and their context-driven temporal expression has not yet been resolved. To address this, [...].

Page 19, paragraph 3:

We sought to assess if structured departures from patterns of spontaneous brain activity that emerge when viewing an ecologically valid sensory experience are functionally meaningful and reproducible. To this end, we focussed on 14 canonical brain networks because such large-scale systems have been consistently implicated at rest and during cognitive, emotional, motor, and perceptual tasks⁶¹⁻⁶⁴. These prior findings suggest that these canonical networks capture core properties of functional brain organisation^{65,66}. This spatial scale of description for whole-brain dynamics has become ubiquitous in the field of cognitive and clinical sciences, facilitating the link between our findings and the existing literature and its future translation to clinical investigations. Other approaches have started from higher dimensional representations⁵⁵ or chosen a data-driven, adaptive dimension reduction⁶⁷. Although we identify rich multi-state dynamics during film viewing, it is likely that the coarse-grained dimension reduction to 14 networks contributed to the restriction of resting-state dynamics largely to two states, less than previously observed^{55,68}. Similarly, defining HMM states across large networks and estimating the model from whole-brain data imposes coordinated, whole-brain transitions. This precludes the identification of hierarchically nested time scales previously reported in film viewing fMRI when HMM models were estimated regionally and not globally^{27,69}. In sum, our choice of the spatial aperture of large functional networks is well tuned to highlight the transition from resting state dynamics to those evoked by movie immersion. Alternative approaches reveal other complex features of these rich neuronal dynamics, including the role of state transitions in scene completion^{70,71}, narrative segmentation⁷² and memory encoding²⁷.

R1.2. For example, it seems that during resting there are fewer states, and less switching. What's the reason for that? Is this a biological reason, e.g. because the brain is "more relaxed" somehow in rest? or is it because the HMM has a harder time in describing the data in the resting-state, due to the data having less structure? In the latter case, the inference of the model would probably put more explanatory power on the movie data, and sort of "give up" on the resting-state.

A. Compared to movie viewing, the resting state is indeed characterised by fewer brain states and a reduced number of switches. Following the reviewer recommendations, we undertook several additional confirmatory analyses. Specifically, we: (i) compared the likelihood of the HMM per time point for both rest and movie (R1.2.1, below), (ii) assessed the summary statistics of HMMs trained on rest and movie data separately

(R1.2.2), and (iii) reproduced our outcome measures with a new HMM inference fitted on movie and rest data of equal size (8 minutes) (R1.2.3).

R1.2.1 One way to investigate this is to compute the likelihood of the model per time point, and see how it differs between rest and movie data (this is the second output argument of the function `hmmfe`, which is in the toolbox the authors are using). A higher likelihood would mean "better explained".

As per suggested, we assessed time-resolved changes in the HMM log-likelihood. Results from the original HMM inference in our manuscript showed that the median (over time) log-likelihood is slightly higher (less negative) at rest compared to movie viewing (paired t-test, $p = 0.03$ for session A and $p = 0.02$ for session B, see new Supplementary Fig. 2 below). These findings argue against the notion that a poor fitting of the HMM on rest data explains the reduced brain dynamism compared to movie viewing.

We also assessed putative differences in the log-likelihood of the HMM when it was separately trained using two 8 minutes sessions or rest and movie data. In session B, results replicate the log-likelihood differences mentioned above. However, in session A we found no statistical differences in the model fit between rest and movie. Results associated with this analysis replicate the original results (see below in R1.2.2), further precluding that a difference in how well the HMM captures rest and movie dynamics can explain the reported findings linked to brain state dynamics.

Supplementary Figure 2. Time-resolved HMM log-likelihood, for both rests and movie data. Grey lines indicate the 95% within-subject confidence intervals (calculated using the optimised Cousineau-Morey

method²). **a** Results obtained by fitting the HMM on 8 minutes rest and 20 minutes of movie data (original analysis presented in the main text). **b** Results from independent HMM inferences performed using 8 minutes of resting-state data and 8 minutes of movie data.

R1.2.2 Another analysis that would help in this regard is to run the HMM separately on rest and movie, and compare summary statistics such as the amount of state switching.

A. We have now run the HMM *separately* on rest and movie. For this analysis, and in line with the following reviewer's comment, we arbitrarily chose an 8-minutes section from the second half of the movie (from time point 271 to 485, see also the new Supplementary Figure 2). Next, we compared the number of brain state switches detected in our initial HMM inference (which included rest and movie) with the new two inferences performed on rest and movie only. To assess the number of switches during movie viewing in the original inference, we only considered the second 8-minute period as described above. We found no significant differences in number of state switches.

Finally, we re-computed the average state path, fractional occupancy, and brain state transitions for HMM inferences performed on 8 min movie/rest data. Results are presented in the new Supplementary Fig. 3 (presented below).

Overall, findings from the confirmatory analyses are in line with what was originally reported.

Supplementary Figure 3. Results from distinct HMM inferences computed on 8 minutes of resting-state and 8 minutes of movie data (two sessions, $N = 14$ participants). The ‘Hungarian algorithm’ was adopted to detect the most representative HMM among 15 inversions^{3,4}. In all panels, brain states are colour-coded according to the legends in panel **b**. **a**: Viterbi State paths for each participant and temporal consistency across participants (between 50 and 100%; bottom trace). **b**: The relative fMRI signal weight onto the 14 canonical networks considered: dorsal and ventral Default Mode Networks (dDMN and vDMN), Precuneus, Anterior Salience Network (ASN), Posterior Salience Network (PSN),

*Left and Right Executive Control Networks (IECN and rECN), Basal Ganglia Network (BGN), Auditory Network (AUD), Primary Visual Network (pVIS), High Visual Network (hVIS), Sensorimotor Network (SMN), Visuospatial Network (VSN), and Language Network (LAN). These are divided into four main groups: DMN (Default Mode Network); SAL (Salience Network); EXEC (Executive Network) and SENS (Sensory Network). **c**: fractional occupancy. **d**: transition probability matrix (left), and state transitions identified by the Network-Based Statistics ($p_{FWE} < 0.05$; one-sample t-test vs null) (right).*

R1.2.3 An important confound that the authors are seemingly not considering is the different amount of data between the two conditions, given that there are more movie data (20min) than resting state data (8min). Do the results hold when only 8min of data are used in the inference? If brain activity is different between these two conditions (which is the case), having much more data for one of the conditions would trivially lead to solutions where the condition with less data uses fewer states.

A. As above, we now inferred the HMM on *concatenated* rest and movie data of similar duration (8 min). For this confirmatory analysis, we ran HMM inferences until we obtained 15 instances which were characterised by the full set of ten brain states (i.e. all states expressed at least once in the data). Next, we selected the most representative inversion using the Hungarian method. The process and the related results are now shown in the new Supplementary figure 4 and Supplementary figure 5 (below). These results replicate what we originally reported using 8 min of rest and 20 minutes of movie data.

Supplementary Figure 4. a: HMM inferences on 8 minute concatenated rest and movie data. The Hungarian algorithm was adopted to detect the most representative HMM inference among 15 inferences^{3,4}. **b:** Brain state dynamics during 8 minutes rest and movie viewing, for each participant (raw) and session (top/bottom). The brain states color-coding is associated to the topology of the state presented in panel c. As per Fig. 1, the temporal consistency across participants, which vary between 50% (chance for the resting state) and 100% (complete consistency) is presented in the bottom panel of each experimental session. **c:** fMRI signal profiles for each of the 10 brain states occurring during resting state and movie viewing. Consistent with results from the primary analyses

(Fig. 1), the relative weighting of each brain state onto each of the 14 canonical networks is considered. The blue-red color bar indicates the relative loading to the average brain states activity.

Analyses of putative differences in fractional occupancy (Supplementary Fig. 5a) and brain state transitions (Supplementary Fig. 5b) confirm a reduced dynamism of brain states at rest compared to movie viewing.

Supplementary Figure 5. Dynamic characteristics of brain states across rest and movie viewing for HMM inference on concatenated rest and movie sessions of 8 minutes. Asterisks indicate statistical significance (t -tests, $p < 0.05$; corrected for multiple comparisons across states). **a**: fractional occupancy (FO) and brain state dwell times. Grey lines show how FO and dwell times are paired within participants. **b**: brain state transition probabilities during 8 minutes rest and 8 minutes movie viewing. Left column: Group averaged transition probability matrices for rest (i) and movie viewing

(iv). Diagonal elements are omitted. Middle column: Top 20% state transitions during rest (ii) and movie viewing (v). Arrow thickness corresponds to the group-averaged probability of that transition. Right column: State transitions having a significantly higher probability of occurring during rest than during movie viewing (iii) and vice versa (vi); identified by the Network-Based Statistics ($p_{FWE} < 0.05$). Each of the colour circles represents a state according to the colour scheme used in Supplementary Fig. 4. The diameter is scaled according to that state's averaged FO (panel a). Each arrow represents a transition.

Based on the above results (A1.2.1, A1.2.2, A1.2.3), the following text has been added in the manuscript:

Page 5, paragraph 1:

The inversion of the HMM from these data yielded ten distinct states (Fig. 1). Confirmatory analyses were performed on 8 minutes of rest and 8 minutes of movie data, with HMM inversions performed on concatenated data as well as performed separately (i.e., movie and rest independently; Supplementary Figures 2-5).

Page 5, paragraph 2:

Brain states are characterised (see Fig. 1) by their distinct fMRI signal loadings onto the 14 canonical brain networks (Supplementary Fig. 1). State 1 is defined by high fMRI signal in most networks. States 5 and 9 show a relatively uniform fMRI signal across networks, whereas State 7 displays a low projection onto all networks (except for the primary visual network). The remaining brain states (2-4, 6, 8 and 10) show idiosyncratic fMRI signal across the 14 brain networks. The fMRI signal defining these brain states load preferentially on one or more specific functional networks, such as those supporting language (state 3), visual-auditory stimuli processing (state 2, 4, 10), and interoception (state 6, 8); see Supplementary Figure 3 for confirmatory analyses on HMM inferences on 8 minutes rest and movie data separately and Supplementary Figure 4 for results on HMM inferences on 8 minutes each of concatenated data).

Page 7, paragraph 1:

As anticipated, movie viewing was associated with greater consistency across participants, relative to rest (Fig. 2; also Supplementary Figs 3, 4, and 6).

Page 13, paragraph 2:

We next quantified the dynamics of the ten brain states by assessing the following measures: (i) fractional occupancy (FO), defined as the proportion of total time spent in each given state; (ii) state dwell time, representing the total amount of time spent in each state, and (iii) state transition probabilities, or the likelihood of specific transitions between distinct brain states (Fig. 4; see Supplementary Figs 3 and Fig. 4 for confirmatory analyses).

R1.3. Also, I would find very interesting to compare inter-session consistency (for each subject separately and averaged across subjects) versus the inter-subject consistency (for each session separately and averaged across sessions). I wonder which one is higher. If the inter-session consistency is higher, that might suggest that there's a movie-related subject-specific neural signature, which I think is intriguing.

A. Qualitatively, the *inter-session* consistency (across sessions within the same subjects) is indeed higher (mean 0.23, SD = 0.04, see the new Supplementary Fig. 7 below) compared to the *inter-subject* consistency (between subjects within the same session; average mean 0.19, SD = 0.04). This difference is confirmed statistically (paired t-test between *inter-session* consistency and the average *inter-subject* consistency: $t_{26} = 2.85$, $p = 0.008$).

The following text has been added on page 8, paragraph 1:

We also assessed the inter-session consistency by calculating the Jaccard index over state visits across session A and session B, averaged over brain states and participants (see Methods). The occurrence of brain states was significantly more consistent during movie viewing than rest (average Jaccard overlap of 0.18 (+/- 0.04) in movies and 0.08 (+/- 0.07) at rest, $p = 0.0020$). This finding is in line with the higher between-session consistency within each participant during movie compared to rest (Supplementary Fig. 6). These results highlight the relatively high (across-session and between subject) consistency of brain state dynamics using a naturalistic stimulus, compared to the unconstrained resting state condition. Formal comparison between inter-session consistency (for each subject separately) and inter-subject consistency (average inter-subject consistency) showed higher inter-session consistency (paired t-test, $t_{26} = 2.85$, $p = 0.008$; see also Supplementary Fig. 7). This result suggests the existence of movie-related participant-specific neural signatures.

The following figure has been added to the supplementary material:

Supplementary Figure 7. Inter-session and inter-subjects consistency of each participant during movie viewing. The inter-subject consistency was calculated using the Jaccard overlap between brain states expression (Viterbi path) averaged across subjects (subject's expression compared to the remaining 13 subjects). The inter-session consistency represents the Jaccard overlap between session A and B in the same subject. Within subject, inter-session consistency is higher than inter-subject consistency (paired t-test, $t_{26} = 2.85$, $p = 0.008$).

Other suggestions:

R1.4. - pp13, line260. "In addition, these visited states were characterised by strong and specific functional network weightings." Do the authors mean functional connectivity? Are

differences in FC reported? Sorry if I've missed this. I think it would be very relevant if that were the case, given the controversy about whether there are meaningful differences in FC or not.

A. We apologies for the lack of clarity. The sentence does not refer to functional connectivity, but to the BOLD signal in each canonical network.

We have now clarified this sentence on page 13, paragraph 2:

Contrary to resting state, movie-induced brain state dynamics occurred with an overall reduction in time spent visiting each specific brain state. Also, these visited brain states were characterised by strong and specific patterns of fMRI signal in canonical functional networks. In other words, movie viewing induced a greater number of briefer brain state visits compared to that during rest [...]

R1.5. - A technical point: could the authors specify upfront the observation model of the HMM? Is it a Gaussian distribution with mean and covariance?

A. We now specify the observation model on page 22, paragraph 5:

The HMM-MAR MATLAB toolbox (<https://github.com/OHBA-analysis/HMM-MAR>) was used to perform Variational Bayes inference on the HMM, using 500 training cycles, according to previously established procedures^{8,56,69}. The HMM assumes that fMRI time series can be described using a dynamic sequence of a limited number of brain states⁸. The total number of states needs to be specified a priori. Previous studies modelling fMRI dynamics in healthy individuals considered between 5 and 12 states^{8,47,67}. In our analysis the HMM input is a 20,860 (14 participants with each 1,490 timepoints) by 14 (average signal from 14 network masks) matrix. In this particular instantiation of the the HMM, each brain state is defined by a multivariate Gaussian distribution, which is described by the mean within each voxel, and covariance between voxels, when each state is active⁸. [...].

Reviewer #2:

In this paper the authors apply a Hidden Markov Modeling approach to characterise brain states during both resting-state and movie-watching data. This was an interesting paper to read, with some very nice and informative figures. There are a number of different analyses presented, which help characterise the discovered states in terms of their functional profiles, impacts on physiological measurements, and stimulus features, and compare dynamics between movies and resting-state as well as across subjects.

Major comments:

R2.1. The framing of this paper is that work on understanding brain dynamics (on the scale of minutes) "has thus far largely adopted resting state paradigms," and that this work extends HMM approaches used for resting-state data to the movie domain. This is a strange framing for this work, since there has been a large amount of recent work characterising brain dynamics in movies, many of which have also used HMMs (and some of which has even been conducted by some of this study's authors). Examples of some of this work that was not discussed in this paper include:

-Applying HMMs to fMRI movie data: Baldassano et al. Neuron 2017; Baldassano et al. JoN 2018; Luke Chang et al. bioRxiv 2018 (cited in the paper, but only regarding the "reverse inference" meta-analysis); James Antony et al. bioRxiv 2020; Heusser, Fitzpatrick, Manning bioRxiv 2018

-Applying HMMs to other task data: Vidaurre et al. Cerebral Cortex 2018

-Another approach for identifying dynamics in movies: Tseng & Poppenk bioRxiv 2019

The authors do not need to cite all of these papers, but should better situate their work relative to this growing literature.

Authors (A). We appreciate the detailed reading constructive feedback of the reviewer. The comment of the reviewer is in line with the first comment of reviewer #1. Accordingly, we have better discussed the literature and repositioned the paper to highlight the importance of our work in highlighting how movie engagement reliably reorganises brain state dynamics compared to a state of rest. The following changes have been made in the manuscript:

Abstract

*Adaptive brain function requires that sensory impressions of the social and natural milieu be dynamically incorporated into intrinsic brain activity, where they are used to update internal models of the world and make predictions about the future. While dynamic switches between brain states have been well characterised in resting state acquisitions, the remodelling of these state transitions by engagement in naturalistic stimuli remains poorly understood. By exploiting the rich perceptual and emotional content of a short movie, we find that the temporal dynamics of brain states, as measured in fMRI, are reshaped from predominantly bistable transitions between two relatively indistinct states *at rest*, toward a sequence of well-defined functional states *during film viewing* whose transitions are temporally aligned to specific features of the movie. The *expression* of these brain states *covaries* with different physiological states (heart rate variability and pupil diameter) and reflects subjectively rated engagement in the film. In sum, a data-driven decoding of brain states reveals the distinct reshaping of functional network expression and reliable state *changes* that accompany *the transition from resting state to perceptual immersion* in an ecologically valid sensory experience. *Juxtaposing resting state with movie viewing acquisitions highlights the influence of naturalistic perceptual streams on brain-state transitions.**

Introduction

Page 4, paragraph 1:

Naturalistic stimuli, such as films⁴ and spoken narratives²⁴ offer the constraint and replicability that resting state acquisitions lack while adding greater ecological validity than traditional task designs^{25,26}. Recent analyses of movie viewing fMRI data using the HMM reveal a hierarchy of timescales, with more frequent state transitions in sensory cortex hierarchically nested within progressively slower transitions in heteromodal regions²⁷. Such multiscale dynamics mirror the statistics of the natural world²⁸ and suggest a remodelling of intrinsic correlations so that their complexity more closely matches the statistical structure of naturalistic perceptual streams^{5,29}.

Comparing unconstrained resting state acquisitions to film viewing holds potential to characterise this process, and hence to investigate the functional significance of transitory brain states. [...]

Discussion

Page 18, paragraph 1:

*The brain is constantly active, with activity across disparate brain regions supporting diverse cognitive functions, which in turn allow appraisal and interaction with the environment^{9,43-45}. Here, we assessed if the temporal dynamics of distinct brain states reflected the sensory, cognitive, emotional and physiological processes underpinning the subjective experience of a theatrical film. In particular, we contrasted resting-state and movie-viewing acquisitions to understand how perceptual immersion in an engaging naturalistic stimulus reshapes whole-brain state transitions. Using a meta-analytic approach, we first found that *movie-viewing elicits a richer repertoire of HMM brain states than those that occur at rest, and which match distinct functional profiles.**

Page 18, paragraph 2:

*However, the functional relevance of these *spontaneous* brain states and their *context-driven temporal expression* has not yet been resolved. To address this, [...].*

Page 19, paragraph 4:

We sought to assess if structured departures from patterns of spontaneous brain activity that emerge when viewing an ecologically valid sensory experience are functionally meaningful and reproducible. To this end, we focussed on 14 canonical brain networks because such large-scale systems have been consistently implicated at rest and during cognitive, emotional, motor, and perceptual tasks⁶¹⁻⁶⁴. These prior findings suggest that these canonical networks capture core properties of functional brain organisation^{65,66}. This spatial scale of description for whole-brain dynamics has become ubiquitous in the field of cognitive and clinical sciences, facilitating the link between our findings and the existing literature and its future translation to clinical investigations. Other approaches have started from higher dimensional representations⁵⁵ or chosen a data-driven, adaptive dimension reduction⁶⁷. Although we identify rich multi-state dynamics during film viewing, it is likely that the coarse-grained dimension reduction to 14 networks contributed to the restriction of resting-state dynamics largely to two states, less than previously observed^{55,68}. Similarly, defining HMM states across large networks and estimating the model from whole-brain data imposes coordinated, whole-brain transitions. This precludes the identification of hierarchically nested time scales previously reported in film viewing fMRI when HMM models were estimated regionally and not globally^{27,69}. In sum, our choice of the spatial aperture of large functional networks is well tuned to highlight the transition from resting state dynamics to those evoked by movie immersion. Alternative approaches reveal other complex features of these rich neuronal dynamics, including the role of state transitions in scene completion^{70,71}, narrative segmentation⁷² and memory encoding²⁷.

R2.2. The HMM fits to the resting-state data are surprisingly poor, consisting primarily of states 5 and 9, which are both essentially constant activity across all ROIs. This is not consistent with the traditional literature on resting-state activity, which usually shows highly structured fluctuations e.g. those used to define the Default Mode Network. The authors claim that this bistability is similar to that described in Hansen et al. 2015, but the conclusions of that paper are quite different - that paper describes bistability in the *connectivity* matrix (not the activity patterns of individual timepoints) and shows substantial structure in which regions co-occur (e.g. see Fig 5 in Hansen et al.). One possibility for why the HMM fit is so poor in the resting-state data is because the HMM was fit to resting-state and movie data concatenated together, and the longer length of the movie data encouraged the model to preferentially identify states that were useful for the movie data only.

A. This is an insightful point. In line with comments from reviewer #1, the reviewer highlights two possible biases in our analyses: (i) poor fit of the HMM to resting-state data because time-series comprised concatenated rest and movie data, (ii) different lengths of data for rest (8 minutes) versus movie (20 minutes). These concerns have been addressed in full by our extensive confirmatory analyses: In brief, re-analysing our resting and movie data separately (not concatenated) AND cropping the length of the movie data to 8 minutes yielded findings that are confirmatory to our original findings, with predominantly bistable states at rest and structured sequences during movie viewing. Please refer to our responses to reviewer #1 (pages 3-10 in this document and additional Supp. Figures 2-5).

Altogether, results from our control analyses confirm that resting-state dynamics are less replicable across experimental sessions and less rich than those that arise during movie engagement. Although our resting state dynamics are largely limited to two states, our results are not incompatible with previous analyses showing highly structured fluctuations in resting state BOLD signal. This is likely because we sought to identify

putative differences in brain dynamics between rest and movie, and hence chose a particular analysis pipeline that may have suppressed some of the idiosyncratic dynamics at rest (see also our response to your third point, below). We made a note regarding this comment in the revised manuscript (page 19, paragraph 1):

Our findings firstly support recent observations that resting state brain dynamics are predominantly bistable, converging with prior models of resting state EEG⁵⁹ and fMRI data⁸. Specifically, we found that two dominant states at rest whose network expressions reflected only subtle modulations of the mean networks activity across all acquisitions. These findings are broadly compatible with results showing structured fluctuations in resting state fMRI data⁶⁰, while also highlighting that such dynamics are significantly less rich and structured than those induced by complex audiovisual stimuli. Indeed, movie viewing imposed richer brain state dynamics that were characterised by distinct functional profiles, with stronger perturbation away from global mean activity – consistent with deeper attractor networks⁶¹.

We also thank the reviewer for pointing out our misguided citation to Hansen et al.. We now use a reference of Vidaurre, PNAS, 2017 and a reference to the seminal work of Fox and colleagues.

R2.3. The representation of the brain state at each timepoint is downsampled to just 14 values, reflecting overall activity in large pre-defined networks. It is unclear why this was necessary, since the HMM could be applied to much higher-dimensional data at the level of individual brain parcels or even individual voxels (though a covariance type other than "full" would need to be selected as the "covtype" in the HMM-MAR toolbox for high-dimensional states). This low-dimensional representation of the brain will mask interesting pattern dynamics at the region level which we know carry information about brain states during movie-watching, e.g. Chen et al. Nature Neuroscience 2017.

From a methodological standpoint, the HMM can indeed be applied to fMRI data with higher (brain parcels) or lower (brain regions) dimensionality. Other approaches allow identification of a data-driven “just right” dimensionality, as per the Chen (2017) paper. The selection of a given dimensionality is intrinsically linked to the question and hypotheses posed by the study. To address this issue more explicitly, we now state (p. 19, paragraph 3):

We sought to assess if structured departures from patterns of spontaneous brain activity that emerge when viewing an ecologically valid sensory experience are functionally meaningful and reproducible. To this end, we focussed on 14 canonical brain networks because such large-scale systems have been consistently implicated at rest and during cognitive, emotional, motor, and perceptual tasks⁶²⁻⁶⁵. These prior findings suggest that these canonical networks capture core properties of functional brain organisation^{66,67}. This spatial scale of description for whole-brain dynamics has become ubiquitous in the field of cognitive and clinical sciences, facilitating the link between our findings and the existing literature and its future translation to clinical investigations. Other approaches have started from higher dimensional representations⁵⁶ or chosen a data-driven, adaptive dimension reduction⁶⁸. Although we identify rich multi-state dynamics during film viewing, it is likely that the coarse-grained dimension reduction to 14 networks contributed to the restriction of resting-state dynamics largely to two states, less than previously observed^{56,69}. Similarly, defining HMM states across large networks and estimating the model from whole-brain data imposes coordinated, whole-brain transitions. This precludes the identification of hierarchically nested time scales previously reported in film viewing fMRI when HMM models were estimated regionally and not globally²⁷. In sum, our choice of the spatial aperture of large functional networks is well tuned to highlight the transition from resting state dynamics to those evoked by movie immersion. Alternative approaches reveal other complex features of these rich neuronal dynamics, including the role of state transitions in scene completion^{70,71}, narrative segmentation⁷² and memory encoding²⁷.

R2.4. The between-subject differences section uses only summary statistics from each subject's neural data, i.e. the overall frequency of occupying each state and the transition matrix between states. However these measures ignore whether two subjects are actually occupying these states *at the same times* during the movie, which seems critical for assessing whether subjects are having a shared "movie experience" - e.g. just knowing that two subjects were both sad during 20% of the movie seems much less important than whether they thought the same scenes were sad. A measure related to the "consistency" calculation in Fig 2 would capture this kind of across-subject alignment.

A. We repeated the analysis linking brain dynamics and behavior (Fig. 7 of the manuscript) using the *Jaccard dissimilarity metric* (i.e., $1 - \text{Jaccard overlap}$) applied to the participants' brain state paths. However, the between-subject dissimilarity in brain state paths was universally high and relatively featureless to extract any meaningful information (see figure below).

Association between brain state paths and behaviour. Across participants dissimilarity in state paths and responses to post-movie questionnaires. In the inter-subject distance matrices each element denote a pair-wise distance (from zero to 1, with 1 representing maximal dis-similarity). The Jaccard dissimilarity metric was used to assess across-participant differences in brain state path. On the other hand, the Euclidean distance was used to infer across-subjects differences in the responses to the post movie questionnaire (session A).

These extra analyses highlight that individual time-locked brain-state paths are quite subject-specific, and its resolution does not match the relatively coarse-grained nature of the questionnaire. Brain state measures with lower granularity, including fractional occupancy and state transitions, are better suited to capture brain-behaviour associations.

We added a following brief summary of these findings to the revised manuscript (p. 15):
We also explored quantitative similarities between the time-locked individual state-paths across participants – i.e. the detailed sequence of states and the precise timing of individual transitions. Such a sequence is a relatively complex path unfolding on a high-dimensional manifold. Not surprisingly, all participants pursued relatively unique paths and the (Jaccard) dissimilarity index was near ceiling for all subject pairs (0.72-0.88). Despite the co-occurrence of states during movie viewing, transition times are not precisely enough aligned to allow a meaningful comparison across participants.

Minor comments:

R2.5. The text description of Fig 1 (lines 102-107) is very confusing, and doesn't seem to correspond to the figure. States are described as belonging to three clusters but it is not clear which states comprise each cluster, states 4 and 9 are described as "mean overall activity" but

the figure describes only state 5 as mean activity (and state 9 as low activity), state 7 is described as low activity but the figure labels state 9 this way (since state 7 does have high visual activity).

A. To improve clarity, we have changed the label in Figure 1 for brain state 9 to ‘relatively uniform networks activity’ (instead of ‘Low brain networks activity’). Furthermore, we have modified the corresponding text of the manuscript on page 5, paragraph 2:

Brain states can be categorised into three broad clusters: (i) high overall activity (e.g., state 1), (ii) mean overall activity (states 4 and 9), and (iii) low overall activity (state 7, Fig. 1). In the context of these broad clusters, a number of states load preferentially on one or more specific functional brain networks. Brain states are characterised (see Fig. 1) by their distinct fMRI signal loadings onto the 14 canonical brain networks (Supplementary Fig. 1). State 1 is defined by high fMRI signal in most networks. States 5 and 9 show a relatively uniform fMRI signal across networks, whereas State 7 displays a low projection onto all networks (except for the primary visual network). The remaining brain states (2-4, 6, 8 and 10) show idiosyncratic fMRI signal across the 14 brain networks. The fMRI signal defining these brain states load preferentially on one or more specific functional networks, such as those supporting language (state 3), visual-auditory stimuli processing (state 2, 4, 10), and interoception (state 6, 8); see Supplementary Figure 3 for confirmatory analyses on HMM inferences on 8 minutes rest and movie data separately and Supplementary Figure 4 for results on HMM inferences on 8 minutes each of concatenated data) ~~and the default mode (state 6).~~

R2.6. One of the annotations compared to movie states is changepoints. However we might expect changepoints to correspond to *switches* between states rather than to a specific state itself – is it possible to assess whether changepoints are related to state switches (between any states or specific pairs of states)?

A. This is an interesting suggestion which we investigated by assessing the link between movie changepoints and brain state switches. Unfortunately, the movie transitions are clustered together, frequently occurring several times a second. For this reason, the temporal resolution of our BOLD-derived brain state transitions do not allow for a meaningful assessment of any putative temporal relationship. Nonetheless, we feel that readers might likely have a similar curiosity and have therefore added the following text and supplementary figure to the revised paper (p. 12):

In principle, it would be interesting to understand whether temporal disruptions in the movie dynamics (e.g. scene changes) coincide with temporal discontinuities in brain states (i.e. state transitions). However, movie change-points tend to be clustered together, frequently occurring several times a second, interspersed by relatively long continuous scenes (Supplementary Fig. 8). The relatively slow temporal resolution of our BOLD-derived brain state transitions thus does not allow for a meaningful assessment of any putative temporal relationships. However, given the frequent occurrence of clustered, bursty dynamics reported in neurophysiological recordings⁴³, the use of rapidly sampled data modes (e.g., M/EEG) could address this question in future studies.

Supplementary Figure 8: Representation of the HMM brain states and the changepoints onsets (vertical black dashed lines) for session A (similar observation for session B). The thin grey lines demarcate the temporal resolution of the MRI acquisition (TR = 2.2 seconds).

Reviewer #3:

Summary: The paper identified ten brain states from fMRI timeseries datasets acquired during 8 minutes of resting state scanning and 20 minutes of watching and listening to a movie. Heart rate, HRV, respiratory rate and volume eyetracking with pupilometry data were also acquired, along with post scan questionnaire ratings. 18 people were analysed from a first session and 14 complete 2 sessions. Brain states were computed using a data-driven Hidden Markov Model framework that, as discussed, 'is not grounded in biophysical models of neural activity' and in which 'statistical fitting of the data that imposes a strict assumption of discretely expressed, not continually mixed, brain states'. The paper uses much reverse inference with respect to interpretation of brain states and highlights the value of movie viewing paradigms that could be extended e.g. to clinical studies as they provide 'rich and reliable' measures of brain dynamics.

Overall I judge that this paper makes an important and valuable contribution to neuroscience literature and understanding of human, building from (ICA) work of Bartels and Zeki 1999, and suggesting a practical (alternative to rsfMRI) approach to quantification and evaluation of brain dynamics measured using fMRI. The sequencing and dwell time of these brain states and the inferred relationship to distinct perceptual and cognitive processes provides novel insights

Authors (A): We appreciate the detailed and helpful feedback of our manuscript.

There are a number of points that I seek clarity about.

R3.1. Typically when describing fMRI brain networks, people look for interregional correlations in fluctuating BOLD signal or concurrent activation across regions. I don't think

these are necessarily the same thing. I therefore would value more guidance regarding how best to think about these brain states and the network activity - activity here is thus presumably the magnitude of the BOLD signal? viz: 'inversion of the HMM from these data yielded ten distinct states. To understand the functional expression of these states, we coded their respective loadings onto each of 14 widely studied canonical brain networks (see Supplementary Fig. 1). The expression of network activity was normalised so that zero corresponds to the average activity of that network across the movies and rest periods. The variability was scaled according to that network's standard deviation. Each network was normalised separately (Methods). What is the rationale for scaling against across the network? Again I would like more guidance as to how I should think about this neural terms, since different regions have quite different neural firing properties

A. The reviewer's intuition is correct, the visualised weights correspond to the (normalised) contribution of each canonical network's average BOLD signal magnitude to the corresponding brain state. Note that the mean-centred scaling is carried out as part of standard HMM preprocessing to allow different participants/conditions to be concatenated. We have made edits to help the reader appreciate the characterisation and meaning of HMM brain states. Specifically, the following changes have been made to the manuscript:

Introduction (page 3, paragraph 2):

Dynamic jumps between discrete brain states, can be modelled using the hidden Markov model (HMM), an analytical framework that posits the existence of distinct states, whose sequential expression yields observed functional imaging data¹². Here, HMM states define spatial patterns of fMRI signal magnitude that recur sporadically in time. The recent application of the HMM to resting state activity has shown that such discrete [...].

Results (page 5, paragraph 1):

To allow for a direct comparison between the states during the different experimental conditions (baseline rest & movie viewing, plus 3-month follow-up rest & movie viewing), we estimated brain states using concatenated timeseries of 14 participants who completed both experimental sessions (Methods). This allowed obtaining a group estimation of brain states for each experimental condition and session. The inversion of the HMM from these data yielded ten distinct states (Fig. 1). Confirmatory analyses were performed on 8 minutes of rest and 8 minutes of movie data, with HMM inversions performed on concatenated data as well as performed separately (i.e., movie and rest independently; Supplementary Figures 2-5). To understand the functional expression of these states, we coded their respective loadings onto each of 14 widely studied canonical brain networks³¹ (Supplementary Fig. 1). The expression of network activity was normalised so that zero corresponds to the average activity of that network across the movies and rest periods. The variability was also scaled according to that network's standard deviation, allowing insights into the balanced representation of changes in fMRI signal magnitude across canonical brain networks in time. Each network was normalised separately (Methods).

Methods (page 24, paragraph 2):

HMM inference yields three different types of output – structural, temporal, and dynamic. The structural output is a heat map overlaid on a template brain for each brain state with the fMRI signal magnitude across the 14 BNs (Fig. 1), together with the functional connectivity/covariance matrices. The variability in the fMRI signal was used to estimate distinct patterns of brain network activity (brain states). The temporal output is the state path [...].

R3.2. It is not always immediately apparent where specific results (brain states) came from (though the figures are very useful) – e.g. HMM analysis of combined resting state and movie datasets over first session, (concatenated over both sessions of the 14 individuals, as indicated above.. why?) or independently in each session. The results text the needs a little more explicit clarification. Related to that is the imbalance in time between the 8 minute resting

state and 20 minute movie. Would the results change if the movie was only 8 minutes long? (particularly the comparative results, and regarding fractional occupancy).

A. This concern resonates with those raised by the first two reviewers. Please refer to our extensive responses to Reviewers #1 and #2. Crucially, in line with comments made by all reviewers, we have performed comprehensive confirmatory analyses ensuring that the reported results are robust (e.g., not driven by different acquisitions time between rest and movie). Edited text and new Supplementary Figures 2-5 are provided above (R1.2; on pages 3-10 of this document): In essence the primary results do not change if the movie data is downsampled to match the 8 window sample length of the resting state, and do not fundamentally differ whether the HMM is estimated independently from the two session types or if the data are concatenated.

We have also clarified that the primary analyses were performed using concatenated 8 minutes rest and 20 minutes movie data (page 23, paragraph 4):

Next, each participant's timeseries were temporally concatenated to a time x BNs matrix (14 participants, each with 215 volumes (8 min) of rest A, 530 volumes (20 min) of first movie viewing, 215 volumes (8 min) for rest B, 530 volumes (20 min) of second movie viewing: 20,860 x 14 BNs matrix). The HMM was then fitted to these temporally concatenated time courses (allowing for covariance between the timeseries) to yield a single set of 10 model parameters (Brain States).

R3.3. The alignment of findings to physiological measures is slightly confusing: The abstract refers to heart rate variability, Table 2 refers to heart rate change not HRV, the discussion refers again to HRV. Both HR and HRV data were available along with respiration (frequency and depth).

A. We have rectified this lack of clarity to refer to HR throughout. We changed the text (abstract, discussion, and method sections) and Table 2 accordingly.

R3.4. It is unclear on what grounds HR(V) and pupil diameter measures were subselected to related to the brain data: why not text all, or look for correspondence between multi-axis body state and brain data.

A. The reason behind the selection of HR and pupil diameter relates to data quality (reliable respiratory data were not available in sufficient numbers of participants). This has now been clarified in the text (page 22, paragraph 4):

Concurrent with functional imaging, (electro-)physiological recordings were also acquired: (i) electrocardiography (ECG) obtained using a Brain Products MR-compatible BrainAmp amplifier (Brain Products GmbH, Gilching, Germany), sampling frequency of 5000 Hz; (ii) respiration obtained from the Scanner's Personal Monitoring Units system, sampling frequency of 50 Hz; and (iii) pupil diameter PD recorded using a MR-compatible EyeLink eyetracker (EyeLink SR Research), sampling frequency of 1000 Hz. Due to the recording quality, only heart rate (extracted from the ECG) and PD were used to link brain states dynamics with physiological changes.

R3.5. Mention is made that heart rate and pupil diameter representing physiological indices of 'arousal'. This may be so in resting state (as might decreased HRV, changes in ventilation) but the issue is more complicated for dynamic audiovisuals aspects of movies. I think the link to physiology needs more consideration – not least as cerebrovascular dynamics have been argued to underpin emergence of some resting state brainwise networks.

A: The reviewer is correct in pointing out that visual inputs are also likely to drive changes in pupil diameter. This prediction is supported by the fact that pupil diameter (associated with emergence of brain state 4: high activity in the visual system linked to face perception) strongly correlated with fluctuations in luminescence (Person's correlation: $r = -0.69$, $p = 10^{-75}$).

The following text has been changed accordingly (Introduction, paragraph 3):

The validity of detected brain states was assessed using cross-session comparisons, movie annotations, the Neurosynth database²⁷, and concurrently recorded physiological indices ~~of arousal~~ including heart rate and pupil diameter.

Page 11, paragraph 1:

*Physiological changes are integral to emotional experiences^{34,35} and naturalistic stimuli are known to evoke reliable physiological changes^{36,37}. We therefore assessed whether the occurrence of a given brain state corresponded to distinct electrophysiological signatures of autonomic function associated with changes in *sensory inputs* and level of arousal, namely heart rate (HR) and pupil diameter (PD)^{38,39}. We observed several significant associations between the occurrence of brain states and fluctuations in both HR and PD (**Table 2**). For example, lower HR was associated with the occurrence of brain state 3 (low dDMN and language network activity), which were in turn characterised by a neutral (i.e. low anxiety and pain) functional profile (**Fig. 3**). *This result supports the link between changes in movie-induced arousal states and the emergence of brain states*. Smaller PD occurred during expression of brain state 4, which was characterised by high visual network activity linked to face perception. *We found a strong negative correlation between PD and scene total luminance ($r = -0.7$, $p = 10^{-87}$), supporting the notion that changes in PD are linked to changes in sensory inputs*. However, larger PD coincided with the occurrence of brain states 1 (high executive, sensory and language) and 2 (high DMN, salience but low executive). In line with high DMN activity, state 2 is functionally linked to high anxiety and pain^{40,41}. *Pupil diameter may therefore also link, at least to some extent, to transient interoceptive mechanisms*. Previous work has shown evoked pupil diameter in the absence of visual stimuli may reflect higher-order cognitive processes, such as updated sensory expectations following surprise⁴², consistent with the association of larger PD and state 1.*

Regarding the contribution of cerebrovascular dynamics to resting state networks, we have added the following caveat to the Discussion (p. 20, paragraph 2):

Finally, the vexed issue of physiological confounds on functional brain networks needs to be acknowledged⁷³. Notably, these effects have most often been identified in resting state acquisitions which, by virtue of their less constrained nature, challenge the disambiguation of nuisance effects from those due to visceral efferents, such as the autonomic correlates of suspense, fear or surprise⁷⁴⁻⁷⁶. Movie viewing may mitigate some of this concern because it comprises structured, emotionally salient material which engenders physiological effects that are correlated with activity in central visceral centres such as the anterior insula³⁵.

R3.6. A little more detail would be useful regarding the resting state (presumed eyes open + fixation cross) for pupillometry but I could not see this stated.

A. The resting state was acquired eye closed. We now clarified this in the manuscript (page 22, paragraph 2):

*The experiment comprised two scanning sessions three months apart. For each session, fMRI data were acquired from participants during an 8-minute *eyes closed* resting-state session, followed by viewing of a 20-min short movie called the "The Butterfly Circus".*

References:

1. Friston, K., Adams, R., Perrinet, L. & Breakspear, M. Perceptions as Hypotheses: Saccades as Experiments. *Front. Psychol.* **3**, (2012).
2. Friston, K. The free-energy principle: a unified brain theory? *Nat. Rev. Neurosci.* **11**, 127 (2010).
3. Grossberg, S. The Link between Brain Learning, Attention, and Consciousness. *Conscious. Cogn.* **8**, 1–44 (1999).
4. Hasson, U. Intersubject Synchronization of Cortical Activity During Natural Vision. *Science* **303**, 1634–1640 (2004).
5. Tononi, G., Sporns, O. & Edelman, G. M. A complexity measure for selective matching of signals by the brain. *Proc. Natl. Acad. Sci.* **93**, 3422–3427 (1996).
6. Allen, E. A. *et al.* Tracking whole-brain connectivity dynamics in the resting state. *Cereb. Cortex N. Y. N 1991* **24**, 663–676 (2014).
7. Hutchison, R. M. *et al.* Dynamic functional connectivity: Promise, issues, and interpretations. *NeuroImage* **80**, 360–378 (2013).
8. Vidaurre, D., Smith, S. M. & Woolrich, M. W. Brain network dynamics are hierarchically organized in time. *Proc. Natl. Acad. Sci.* **114**, 12827–12832 (2017).
9. Zalesky, A., Fornito, A., Cocchi, L., Gollo, L. L. & Breakspear, M. Time-resolved resting-state brain networks. *Proc. Natl. Acad. Sci.* **111**, 10341–10346 (2014).
10. Damoiseaux, J. S. *et al.* Consistent resting-state networks across healthy subjects. *Proc. Natl. Acad. Sci. U. S. A.* **103**, 13848–13853 (2006).
11. Liégeois, R. *et al.* Resting brain dynamics at different timescales capture distinct aspects of human behavior. *Nat. Commun.* **10**, 2317 (2019).
12. Hansen, E. C. A., Battaglia, D., Spiegler, A., Deco, G. & Jirsa, V. K. Functional connectivity dynamics: Modeling the switching behavior of the resting state. *NeuroImage* **105**, 525–535 (2015).
13. Rabinovich, M. I., Huerta, R., Varona, P. & Afraimovich, V. S. Transient Cognitive Dynamics, Metastability, and Decision Making. *PLOS Comput. Biol.* **4**, e1000072 (2008).
14. Roberts, J. A. *et al.* Metastable brain waves. *Nat. Commun.* **10**, 1056 (2019).
15. Tognoli, E. & Kelso, J. A. S. The Metastable Brain. *Neuron* **81**, 35–48 (2014).

16. Cocchi, L., Gollo, L. L., Zalesky, A. & Breakspear, M. Criticality in the brain: A synthesis of neurobiology, models and cognition. *Prog. Neurobiol.* **158**, 132–152 (2017).
17. Freyer, F., Roberts, J. A., Ritter, P. & Breakspear, M. A Canonical Model of Multistability and Scale-Invariance in Biological Systems. *PLoS Comput. Biol.* **8**, e1002634 (2012).
18. Parkes, L., Fulcher, B., Yücel, M. & Fornito, A. An evaluation of the efficacy, reliability, and sensitivity of motion correction strategies for resting-state functional MRI. *NeuroImage* **171**, 415–436 (2018).
19. Power, J. D., Barnes, K. A., Snyder, A. Z., Schlaggar, B. L. & Petersen, S. E. Spurious but systematic correlations in functional connectivity MRI networks arise from subject motion. *NeuroImage* **59**, 2142–2154 (2012).
20. Cole, M. W., Bassett, D. S., Power, J. D., Braver, T. S. & Petersen, S. E. Intrinsic and Task-Evoked Network Architectures of the Human Brain. *Neuron* **83**, 238–251 (2014).
21. Gratton, C., Laumann, T. O., Gordon, E. M., Adeyemo, B. & Petersen, S. E. Evidence for Two Independent Factors that Modify Brain Networks to Meet Task Goals. *Cell Rep.* **17**, 1276–1288 (2016).
22. Gratton, C. *et al.* Functional Brain Networks Are Dominated by Stable Group and Individual Factors, Not Cognitive or Daily Variation. *Neuron* **98**, 439-452.e5 (2018).
23. Hasson, U., Yang, E., Vallines, I., Heeger, D. J. & Rubin, N. A Hierarchy of Temporal Receptive Windows in Human Cortex. *J. Neurosci.* **28**, 2539–2550 (2008).
24. Honey, C. J. *et al.* Slow Cortical Dynamics and the Accumulation of Information over Long Timescales. *Neuron* **76**, 423–434 (2012).
25. Hasson, U. *et al.* Neurocinematics: The Neuroscience of Film. *Projections* **2**, 1–26 (2008).
26. Sonkusare, S., Breakspear, M. & Guo, C. Naturalistic Stimuli in Neuroscience: Critically Acclaimed. *Trends Cogn. Sci.* **23**, 699–714 (2019).
27. Baldassano, C. *et al.* Discovering Event Structure in Continuous Narrative Perception and Memory. *Neuron* **95**, 709-721.e5 (2017).
28. Simoncelli, E. P. & Olshausen, B. A. Natural Image Statistics and Neural Representation. *Annu. Rev. Neurosci.* **24**, 1193–1216 (2001).
29. Olshausen, B. A. & Lewicki, M. S. What natural scene statistics can tell us about cortical representation. *New Vis. Neurosci.* 1247–1262 (2014).
30. Yarkoni, T., Poldrack, R. A., Nichols, T. E., Van Essen, D. C. & Wager, T. D. Large-scale automated synthesis of human functional neuroimaging data. *Nat. Methods* **8**, 665–670 (2011).

31. Shirer, W. R., Ryali, S., Rykhlevskaia, E., Menon, V. & Greicius, M. D. Decoding subject-driven cognitive states with whole-brain connectivity patterns. *Cereb. Cortex N. Y. N 1991* **22**, 158–165 (2012).
32. Chang, L. J., Yarkoni, T., Khaw, M. W. & Sanfey, A. G. Decoding the Role of the Insula in Human Cognition: Functional Parcellation and Large-Scale Reverse Inference. *Cereb. Cortex* **23**, 739–749 (2013).
33. Critchley, H. D. & Harrison, N. A. Visceral Influences on Brain and Behavior. *Neuron* **77**, 624–638 (2013).
34. Seth, A. K. Interoceptive inference, emotion, and the embodied self. *Trends Cogn. Sci.* **17**, 565–573 (2013).
35. Gross, J. J. & Levenson, R. W. Emotion elicitation using films. *Cogn. Emot.* **9**, 87–108 (1995).
36. Nguyen, V. T., Breakspear, M., Hu, X. & Guo, C. C. The integration of the internal and external milieu in the insula during dynamic emotional experiences. *NeuroImage* **124**, 455–463 (2016).
37. Azarbarzin, A., Ostrowski, M., Hanly, P. & Younes, M. Relationship between Arousal Intensity and Heart Rate Response to Arousal. *Sleep* **37**, 645–653 (2014).
38. Joshi, S., Li, Y., Kalwani, R. M. & Gold, J. I. Relationships between Pupil Diameter and Neuronal Activity in the Locus Coeruleus, Colliculi, and Cingulate Cortex. *Neuron* **89**, 221–234 (2016).
39. Xu, J. *et al.* Anxious brain networks: A coordinate-based activation likelihood estimation meta-analysis of resting-state functional connectivity studies in anxiety. *Neurosci. Biobehav. Rev.* **96**, 21–30 (2019).
40. Baliki, M. N., Geha, P. Y., Apkarian, A. V. & Chialvo, D. R. Beyond Feeling: Chronic Pain Hurts the Brain, Disrupting the Default-Mode Network Dynamics. *J. Neurosci.* **28**, 1398–1403 (2008).
41. Filipowicz, A. L., Glaze, C. M., Kable, J. W. & Gold, J. I. Pupil diameter encodes the idiosyncratic, cognitive complexity of belief updating. *eLife* **9**, e57872 (2020).
42. Roberts, J. A., Boonstra, T. W. & Breakspear, M. The heavy tail of the human brain. *Curr. Opin. Neurobiol.* **31**, 164–172 (2015).
43. Kriegeskorte, N., Mur, M. & Bandettini, P. A. Representational similarity analysis - connecting the branches of systems neuroscience. *Front. Syst. Neurosci.* **2**, (2008).
44. Baar, J. M. van, Chang, L. J. & Sanfey, A. G. The computational and neural substrates of moral strategies in social decision-making. *Nat. Commun.* **10**, 1483 (2019).

45. Chen, P.-H. A., Jolly, E., Cheong, J. H. & Chang, L. J. Inter-subject representational similarity analysis reveals individual variations in affective experience when watching erotic movies. *bioRxiv* 726570 (2019) doi:10.1101/726570.
46. Finn, E. S. *et al.* Idiosynchrony: From shared responses to individual differences during naturalistic neuroimaging. *NeuroImage* **215**, 116828 (2020).
47. Cocchi, L. *et al.* Neural decoding of visual stimuli varies with fluctuations in global network efficiency. *Hum. Brain Mapp.* **38**, 3069–3080 (2017).
48. Raichle, M. E. The Brain's Dark Energy. *Science* **314**, 1249–1250 (2006).
49. Shine, J. M. *et al.* The Dynamics of Functional Brain Networks: Integrated Network States during Cognitive Task Performance. *Neuron* **92**, 544–554 (2016).
50. Bartels, A. & Zeki, S. Functional brain mapping during free viewing of natural scenes. *Hum. Brain Mapp.* **21**, 75–85 (2004).
51. Baker, A. P. *et al.* Fast transient networks in spontaneous human brain activity. *Elife* **3**, e01867 (2014).
52. Vidaurre, D. *et al.* Spontaneous cortical activity transiently organises into frequency specific phase-coupling networks. *Nat. Commun.* **9**, 2987 (2018).
53. Bolton, T. A. W., Tarun, A., Sterpenich, V., Schwartz, S. & Van De Ville, D. Interactions Between Large-Scale Functional Brain Networks are Captured by Sparse Coupled HMMs. *IEEE Trans. Med. Imaging* **37**, 230–240 (2018).
54. Nguyen, V. T. *et al.* Distinct Cerebellar Contributions to Cognitive-Perceptual Dynamics During Natural Viewing. *Cereb. Cortex* **27**, 5652–5662 (2017).
55. Vidaurre, D. *et al.* Discovering dynamic brain networks from big data in rest and task. *NeuroImage* **180**, 646–656 (2018).
56. Vidaurre, D., Myers, N. E., Stokes, M., Nobre, A. C. & Woolrich, M. W. Temporally Unconstrained Decoding Reveals Consistent but Time-Varying Stages of Stimulus Processing. *Cereb. Cortex* **29**, 863–874 (2019).
57. Vidaurre, D. *et al.* Spectrally resolved fast transient brain states in electrophysiological data. *NeuroImage* **126**, 81–95 (2016).
58. Freyer, F., Aquino, K., Robinson, P. A., Ritter, P. & Breakspear, M. Bistability and Non-Gaussian Fluctuations in Spontaneous Cortical Activity. *J. Neurosci.* **29**, 8512–8524 (2009).
59. Fox, M. D. *et al.* The human brain is intrinsically organized into dynamic, anticorrelated functional networks. *Proc. Natl. Acad. Sci.* **102**, 9673–9678 (2005).

60. Breakspear, M. Dynamic models of large-scale brain activity. *Nat. Neurosci.* **20**, 340–352 (2017).
61. Fox, M. D., Snyder, A. Z., Zacks, J. M. & Raichle, M. E. Coherent spontaneous activity accounts for trial-to-trial variability in human evoked brain responses. *Nat. Neurosci.* **9**, 23–25 (2006).
62. Vincent, J. L. *et al.* Intrinsic functional architecture in the anaesthetized monkey brain. *Nature* **447**, 83–86 (2007).
63. Smith, S. M. *et al.* Correspondence of the brain's functional architecture during activation and rest. *Proc. Natl. Acad. Sci.* **106**, 13040–13045 (2009).
64. Cocchi, L. *et al.* Complexity in Relational Processing Predicts Changes in Functional Brain Network Dynamics. *Cereb. Cortex* **24**, 2283–2296 (2014).
65. Fox, M. D. & Raichle, M. E. Spontaneous fluctuations in brain activity observed with functional magnetic resonance imaging. *Nat. Rev. Neurosci.* **8**, 700–711 (2007).
66. Fornito, A., Zalesky, A. & Bullmore, E. T. Network scaling effects in graph analytic studies of human resting-state fMRI data. *Front. Syst. Neurosci.* **4**, (2010).
67. Chen, J. *et al.* Shared memories reveal shared structure in neural activity across individuals. *Nat. Neurosci.* **20**, 115–125 (2017).
68. Kottaram, A. *et al.* Brain network dynamics in schizophrenia: Reduced dynamism of the default mode network. *Hum. Brain Mapp.* (2019) doi:10.1002/hbm.24519.
69. Chang, L. J. *et al.* Endogenous variation in ventromedial prefrontal cortex state dynamics during naturalistic viewing reflects affective experience. *bioRxiv* 487892 (2018).
70. Antony, J. W. *et al.* Behavioral, physiological, and neural signatures of surprise during naturalistic sports viewing. *bioRxiv* 2020.03.26.008714 (2020) doi:10.1101/2020.03.26.008714.
71. Heusser, A. C., Fitzpatrick, P. C. & Manning, J. R. How is experience transformed into memory? *bioRxiv* 409987 (2018) doi:10.1101/409987.
72. Baldassano, C., Hasson, U. & Norman, K. A. Representation of Real-World Event Schemas during Narrative Perception. *J. Neurosci.* **38**, 9689–9699 (2018).
73. Chen, J. E. *et al.* Resting-state “physiological networks”. *NeuroImage* **213**, 116707 (2020).
74. Chang, C. *et al.* Association between heart rate variability and fluctuations in resting-state functional connectivity. *NeuroImage* **68**, 93–104 (2013).
75. Laumann, T. O. *et al.* On the Stability of BOLD fMRI Correlations. *Cereb. Cortex* **27**, 4719–4732 (2017).

76. Uddin, L. Q. Bring the Noise: Reconceptualizing Spontaneous Neural Activity. *Trends Cogn. Sci.* (2020) doi:10.1016/j.tics.2020.06.003.
77. Conwell, K. *et al.* Test-retest variability of resting-state networks in healthy aging and prodromal Alzheimer's disease. *NeuroImage Clin.* **19**, 948–962 (2018).
78. Zhang, C., Baum, S. A., Adduru, V. R., Biswal, B. B. & Michael, A. M. Test-retest reliability of dynamic functional connectivity in resting state fMRI. *NeuroImage* **183**, 907–918 (2018).
79. Wang, J. *et al.* Test-retest reliability of functional connectivity networks during naturalistic fMRI paradigms: Test-Retest Reliability of Naturalistic fMRI. *Hum. Brain Mapp.* **38**, 2226–2241 (2017).
80. Esteban, O. *et al.* fMRIPrep: a robust preprocessing pipeline for functional MRI. *Nat. Methods* **16**, 111 (2019).
81. Gorgolewski, K. *et al.* Nipype: A Flexible, Lightweight and Extensible Neuroimaging Data Processing Framework in Python. *Front. Neuroinformatics* **5**, (2011).
82. Pruim, R. H. R. *et al.* ICA-AROMA: A robust ICA-based strategy for removing motion artifacts from fMRI data. *NeuroImage* **112**, 267–277 (2015).
83. Hearne, L. J., Cocchi, L., Zalesky, A. & Mattingley, J. B. Reconfiguration of Brain Network Architectures between Resting-State and Complexity-Dependent Cognitive Reasoning. *J. Neurosci.* **37**, 8399–8411 (2017).
84. Sun, F. T., Miller, L. M. & D'Esposito, M. Measuring interregional functional connectivity using coherence and partial coherence analyses of fMRI data. *NeuroImage* **21**, 647–658 (2004).
85. Niazy, R. K., Beckmann, C. F., Lannetti, G. D., Brady, J. M. & Smith, S. M. Removal of FMRI environment artifacts from EEG data using optimal basis sets. *NeuroImage* **28**, 720–737 (2005).
86. Kasper, L. *et al.* The PhysIO Toolbox for Modeling Physiological Noise in fMRI Data. *J. Neurosci. Methods* **276**, 56–72 (2017).
87. Zalesky, A., Fornito, A. & Bullmore, E. T. Network-based statistic: identifying differences in brain networks. *Neuroimage* **53**, 1197–1207 (2010).
88. Szymkiewicz, D. Une contribution statistique à la géographie floristique. *Acta Soc. Bot. Pol.* **11**, 249–265 (1934).

Reviewers' Comments:

Reviewer #1:

Remarks to the Author:

I feel my concerns and the other reviewers' were addressed carefully, so at this stage my position is to endorse publication.

There is only one minor technical point left, which I think was not sufficiently well explained (or, at least, that I couldn't understand). This is about the use of the Hungarian algorithm to find the most representative solution. The Hungarian algorithm, to the extent of my knowledge, is used just to pair the states between 2 HMM runs. So, once the states are matched somehow across all HMM runs (is it done in just one go of the Hungarian algorithm or iteratively?), how is the most representative solution chosen?

Reviewer #2:

Remarks to the Author:

The authors have made substantial revisions which have largely addressed my concerns.

My only comment is on point R2.4 in the rebuttal, regarding the between-subjects differences. The authors state that a dissimilarity metric based on the temporal alignment of brain states across subjects cannot be used for pairwise comparisons across subjects, since the dissimilarity is "uniformly high" (similarities of 0.12-0.28), reflecting "relatively unique paths" for all subjects. However, when discussing what I believe is the same inter-subject similarity measure on page 8 of the revised manuscript, the authors describe a Jaccard similarity of 0.18 as "relatively high" (compared to rest) and "high, reaching 100% during the viewing of specific movie scenes." I agree with the authors' statements in the paper, that across-subject similarities of 0.12-0.28 reflect a meaningful level of alignment across subjects, and since these similarities are above the noise floor I don't understand why they are considered "relatively featureless" in the rebuttal (with a different colormap in the matrix in the rebuttal figure would show more variability). It doesn't have a large impact on the authors' conclusions either way, but it would be helpful for readers to know if the statepath dissimilarity correlates with the questionnaire distances. It may also be useful for the authors to stress that this is a questionnaire about subjective emotional response and not a quiz on the content of the movie, since this second type of memory test has been previously associated with inter-subject correlation during perception (Hasson et al. Neuron 2008).

Reviewer #3:

Remarks to the Author:

This is a second review of the paper. The authors have addressed earlier comments with additional analysis including downsampling to 8 minutes.

Data on physiology (heart rate and pupil) is clarified, including recognition that much of pupil response is luminance driven. This does not exclude luminance conveying visual salience effecting arousal affecting brain network transitions that would be absent on closed eye resting state conditions.

Heart rate variability is only referred to in the abstract; presumably an oversight that needs correcting?

I remain unsure how the physiological measures relate the rest of the paper. Table 2 links the measure software heart rate to state 3 and pupil to states 1, 2 & 4. Is this incidental or exploratory or, if pupils index luminance, do we infer many of the brain states themselves reflect low level aspects of

visuosensory stimulation rather than cognition.

If predictable switching between states is a critical aspect of the paper, should not a question be whether physical change in physiological signals relate to state shifts.

These are minor points that could be addressed in a line or two of the discussion

Referee responses, 2nd Revision of NCOMMS-20-10106

We have revised our manuscript according to the final comments provided by the three referees and the editorial requests. The editorial requests are handled in the **Editorial_Responses.docx** document.

The Referee's comments and our responses are presented below. We give a point by point response to the comments, together with the edits to the text which are *highlighted in italic blue text*.

Yours sincerely,

Johan van der Meer, Michael Breakspear and Luca Cocchi [on behalf of all authors]

Reviewer #1

I feel my concerns and the other reviewers' were addressed carefully, so at this stage my position is to endorse publication.

There is only one minor technical point left, which I think was not sufficiently well explained (or, at least, that I couldn't understand). This is about the use of the Hungarian algorithm to find the most representative solution. The Hungarian algorithm, to the extent of my knowledge, is used just to pair the states between 2 HMM runs. So, once the states are matched somehow across all HMM runs (is it done in just one go of the Hungarian algorithm or iteratively?), how is the most representative solution chosen?

R1.1: We apologise for the lack of clarity. Accordingly, we have revised the figure legends of Supplementary Fig. 3 and Supplementary Fig. 4:

Supplementary material page 5-6:

Supplementary Figure 3. Results from distinct HMM inversions computed on 8 minutes of resting-state and 8 minutes of movie data (two sessions, N = 14 participants). *In order to detect the most representative HMM inference among the 15 inversions, we used the Hungarian method^{3,4} on the emission probabilities to pair the brain states of each inversion with those of all other inversions. This procedure resulted in 14 values (Pearson's correlations) per inversion, which were then averaged. The inversion with the highest average correlation value was considered as the most representative and used for subsequent analyses. ~~The 'Hungarian algorithm' was adopted to detect the most representative HMM among 15 inversions^{3,4}.~~ In all panels, brain states are colour-coded according to the legends in panel b. a: Viterbi State paths for each participant and temporal consistency across participants (between 50 and 100%; bottom trace). [...]*

Supplementary material page 7:

Supplementary Figure 4. a: HMM inferences on 8 minute concatenated rest and movie data. *We estimated the most representative HMM inversion between a total of 15 inversions using the Hungarian method (see the legend of Supplementary Figure 3 for details). ~~The Hungarian algorithm was adopted to detect the most representative HMM inference among 15 inferences^{3,4}.~~ b: Brain state dynamics during 8 minutes rest and movie viewing, for each participant (raw) and session (top/bottom). The brain states color-coding is associated to the topology of the state presented in panel c. [...]*

Reviewer #2

The authors have made substantial revisions which have largely addressed my concerns.

My only comment is on point R2.4 in the rebuttal, regarding the between-subjects differences. The authors state that a dissimilarity metric based on the temporal alignment of brain states across subjects cannot be used for pairwise comparisons across subjects, since the dissimilarity is "uniformly high" (similarities of 0.12-0.28), reflecting "relatively unique paths" for all subjects. However, when discussing what I believe is the same inter-subject similarity measure on page 8 of the revised manuscript, the authors describe a Jaccard similarity of 0.18 as "relatively high" (compared to rest) and "high, reaching 100% during the viewing of specific movie scenes." I agree with the authors' statements in the paper, that across-subject similarities of 0.12-0.28 reflect a meaningful level of alignment across subjects, and since these similarities are above the noise floor I don't understand why they are considered "relatively featureless" in the rebuttal (with a different colormap in the matrix in

the rebuttal figure would show more variability). It doesn't have a large impact on the authors' conclusions either way, but it would be helpful for readers to know if the statepath dissimilarity correlates with the questionnaire distances. It may also be useful for the authors to stress that this is a questionnaire about subjective emotional response and not a quiz on the content of the movie, since this second type of memory test has been previously associated with inter-subject correlation during perception (Hasson et al. Neuron 2008).

R2.1: We agree with the reviewer comment. Accordingly, we have reworded the text that introduced the inconsistency in the manuscript (p. 10):

We also explored quantitative similarities between the time-locked individual state-paths across participants – that is, the detailed sequence of states and the precise timing of individual transitions. Although such a sequence ~~Such a sequence~~ is a relatively complex path unfolding on a high-dimensional manifold, ~~the. Not surprisingly, all participants pursued relatively unique paths and the~~ (Jaccard) dissimilarity index ~~was near ceiling~~ for all subject pairs ranged from (0.72-0.88), a moderate level of agreement. That is, ~~Despite~~ despite the misalignment in the precise timing of the state transition times there is still a meaningful commonality of state paths ~~co-occurrence of similar states during movie viewing, transition times are not precisely enough aligned to allow a meaningful comparison~~ across participants.

Additionally, we now undertake comparisons between statepath dissimilarity and questionnaire distances, adding these to the Results section on P. 11:

[...] Differences in FO and questionnaire representation were positively correlated ($r = 0.174$, $p = 0.031$). We also observed a positive correlation between the between-subject distance in the pattern of brain state transition and answers to the post-movie questionnaire ($r = 0.182$ $p = 0.034$) (Fig. 7b). There was no statistically significant association between the state path dissimilarity and the movie ratings ($r = -0.09$ $p = 0.27$).

We describe the extra correlation analysis in the Methods section on P. 20-21:

We used the inter-subject representational similarity analysis (IS-RSA) to assess how brain and behavioural data are represented in a group sample. ~~Three~~ Four inter-subject distance matrices (each 18×17) were constructed for representations of brain dynamics (FO and state transition), brain state Viterbi path, and movie impressions (Fig. 7). To calculate inter-subject distances for movie impression (session A), the Euclidean distance of questionnaire ratings between each possible pair of participants was measured producing the 18 (participants) \times 17 matrix. For the FO representation, for every possible pair of participants, the correlation between the 10 FO values (one for each state) was calculated to produce the inter-subject distance matrix. ~~Finally, also for~~ For the state transition representation, for every possible pair of participants, the correlation between the state transition matrices (10 states \times 9 transitions to another state) was calculated to produce the inter-subject distance matrix. Finally, for the brain state (Viterbi) path, the Jaccard dissimilarity index (1-Jaccard index) was used. In this way we generated a single representation for movie impressions, two representations for brain state dynamics (one for FO and one for state transitions) and one representation for brain state paths. [...]

Finally, we also better discuss how our results relate to previous findings (P. 13):

[...] This finding also highlights that discrete snapshots of patterns of brain activity isolated, using relatively simple low dimensional representations, can capture the complex brain dynamics underpinning our rich subjective experience. Previous work has shown that the recall of a movie's content is associated with inter-subject correlation in fMRI timeseries during movie viewing⁴⁹. Our work adds to this by showing that recall of emotional responses are also linked to common brain state transitions.

Reviewer #3

This is a second review of the paper. The authors have addressed earlier comments with addition analysis including downsampling to 8 minutes. Data on physiology (heart rate and pupil) is clarified, including recognition that much of pupil response is luminance driven. This does not exclude luminance conveying visual salience effecting arousal affecting brain network transitions that would be absent on closed eye resting state conditions. Heart rate variability is only referred to in the abstract; presumably an oversight that needs correcting?

R3.2: Thank you for noticing this oversight, we revised the abstract accordingly:

[...] The expression of these brain states covaries with different physiological states—~~heart rate variability and pupil diameter~~—and reflects subjectively rated engagement in the film. [...]

I remain unsure how the physiological measures relate the rest of the paper. Table 2 links the measure software heart rate to state 3 and pupil to states 1, 2 & 4. Is this incidental or exploratory or, if pupils index luminance, do we infer many of the brain states themselves reflect low level aspects of visuosensory stimulation rather than cognition. If predictable switching between states is a critical aspect of the paper, should not a question be whether physical change in physiological signals relate to state shifts.

These are minor points that could be addressed in a line or two of the discussion

R.3: As requested, we added the following section to the discussion (P. 19):

[...] These state transitions were temporally aligned with the narrative structure of the movie and displayed a close association with corresponding sensory, perceptual, and emotional content, and mirrored changes in HR variability and PD. The link between physiological changes and brain state dynamics suggest that the sensory properties of a movie, as well as the content of its narrative, could be manipulated to evoke discrete brain processes. These findings motivate further investigations into the use of structured naturalistic stimuli to induce sequences of brain states underpinning a broad range of sensory and cognitive processes.